# MAPK signaling promotes axonal degeneration by speeding the turnover of the axonal maintenance factor NMNAT2

Lauren J Walker[1], Daniel W Summers[2], Yo Sasaki[2], EJ Brace[1], Jeffrey Milbrandt[2,3], Aaron DiAntonio[1,3]*

[1]Department of Developmental Biology, Washington University Medical School, Saint Louis, United States; [2]Department of Genetics, Washington University Medical School, Saint Louis, United States; [3]Hope Center for Neurological Disorders, Saint Louis, United States

**Abstract** Injury-induced (Wallerian) axonal degeneration is regulated via the opposing actions of pro-degenerative factors such as SARM1 and a MAPK signal and pro-survival factors, the most important of which is the $NAD^+$ biosynthetic enzyme NMNAT2 that inhibits activation of the SARM1 pathway. Here we investigate the mechanism by which MAPK signaling facilitates axonal degeneration. We show that MAPK signaling promotes the turnover of the axonal survival factor NMNAT2 in cultured mammalian neurons as well as the *Drosophila* ortholog dNMNAT in motoneurons. The increased levels of NMNAT2 are required for the axonal protection caused by loss of MAPK signaling. Regulation of NMNAT2 by MAPK signaling does not require SARM1, and so cannot be downstream of SARM1. Hence, pro-degenerative MAPK signaling functions upstream of SARM1 by limiting the levels of the essential axonal survival factor NMNAT2 to promote injury-dependent SARM1 activation. These findings are consistent with a linear molecular pathway for the axonal degeneration program.

*For correspondence: diantonio@ wustl.edu

**Competing interests:** The authors declare that no competing interests exist.

## Introduction

Axon loss is a hallmark of many neurological disorders. Developing methods to maintain axons and preserve neural circuit integrity following injury or disease will require a mechanistic understanding of the axonal degeneration program (*Conforti et al., 2014*). Injury-induced axonal degeneration is a regulated process of subcellular self-destruction that is controlled by the counteracting functions of axon survival and axon degenerative proteins. Over the past few years, essential survival and degenerative proteins have been identified (*Gerdts et al., 2016*). Now the key challenge is to delineate the relationship among these factors. Here we study this interplay, and present a unified model that incorporates major axon survival and degenerative proteins into a linear pathway.

Axon survival factors inhibit axon degeneration after injury. Overexpression of these factors delays or prevents injury-dependent axon degeneration, and loss of these factors accelerates or induces axon degeneration. NMNAT2, an $NAD^+$ biosynthetic enzyme, is an essential axon survival factor whose loss is sufficient to trigger axon degeneration (*Gilley and Coleman, 2010*). NMNAT2 has a very short half-life, and maintenance of NMNAT2 in the axon requires continuous anterograde transport. Axonal injury blocks axon transport, causing local NMNAT2 levels to fall and axon degeneration to initiate. Factors that slow the degradation of NMNAT2 profoundly delay axon degeneration after injury (*Babetto et al., 2013*; *Milde et al., 2013a, 2013b*; *Xiong et al., 2012*). SCG10, a microtubule regulating protein also known as Stathmin 2, is a less potent axonal survival factor whose loss speeds degeneration after injury but does not trigger degeneration in the absence of

injury (*Shin et al., 2012*). Like NMNAT2, SCG10 is a labile protein that is delivered to the axon via anterograde transport, and slowing its degradation protects axons (*Shin et al., 2012*).

Axon degeneration factors promote degeneration and consequently loss-of-function mutants in these factors slow or block axon degeneration after injury. The MAP3K dual leucine-zipper kinase (DLK) and its downstream MAPK JNK were the first axon degeneration factors identified (*Miller et al., 2009*). This MAPK pathway was thought to play an ancillary role in the axonal degeneration program (*Wang et al., 2012*) because loss of DLK only leads to a modest slowing of degeneration. Recently Yang and colleagues found redundancy at each level of the MAPK cascade, showing that loss of all of the relevant triple, double, or single kinases leads to a long-lasting block in axonal degeneration, and hence demonstrating that MAPK signaling is essential to promote local axon degeneration (*Yang et al., 2015*). SARM1 is another essential axon degeneration factor, and likely functions as the central executioner of the degenerative cascade. Loss of SARM1 protects axons after sciatic nerve injury, traumatic brain injury, or treatment with the chemotherapeutic drug vincristine in vivo in mouse (*Geisler et al., 2016*; *Gerdts et al., 2013*; *Henninger et al., 2016*; *Osterloh et al., 2012*) and activated versions of SARM1 are sufficient to trigger axon degeneration (*Gerdts et al., 2013*, *2015*; *Yang et al., 2015*).

Four recent studies have identified mechanistic links among these axon survival and degeneration programs. Two studies suggest that loss of NMNAT2 is a likely trigger for SARM1 activation. Genetic analysis shows that knockdown of NMNAT2 in cultured neurons triggers SARM1-dependent axon degeneration, and the embryonic lethality of NMNAT2 knockout mice is fully rescued by the simultaneous loss of SARM1 (*Gilley et al., 2015*). Biochemical studies demonstrate that expression of NMNAT enzymes blocks axon degeneration, not by increasing $NAD^+$ synthesis, but rather by blocking SARM1-dependent $NAD^+$ depletion (*Sasaki et al., 2016*). These results support the model that loss of NMNAT2 activates SARM1 to induce degeneration. How does activated SARM1 cause axon loss? SARM1 is a TIR-domain containing protein, and dimerization of the SARM1-TIR domain is sufficient to trigger axon degeneration (*Gerdts et al., 2015*; *Yang et al., 2015*). Using this system, Gerdts et al. demonstrated that activated SARM1 triggers the rapid degradation of $NAD^+$ and the subsequent loss of ATP, inducing a metabolic catastrophe and axon loss (*Gerdts et al., 2015*). Yang et al. showed that dimerizing the SARM1-TIR domains activates MAPK signaling, that injury induces a SARM1-dependent activation of the MAPK signaling pathway, and that MAPK signaling is required for ATP loss after injury (*Yang et al., 2015*). Hence activated SARM1 can induce both $NAD^+$ depletion and MAPK activation, but the relationship between these two activities is unknown.

We hypothesized that SARM1-dependent MAPK signaling was either a cause or consequence of $NAD^+$ depletion, so we investigated the relationship among SARM1 activation, MAPK activation, $NAD^+$ depletion, and axon degeneration. We find that MAPK signaling is required for injury-dependent $NAD^+$ depletion and axonal degeneration, but surprisingly not for $NAD^+$ depletion or axon degeneration induced by activated SARM1. This suggests that MAPK signaling could function upstream of SARM1 to promote injury-dependent activation of SARM1. We previously demonstrated that MAPK signaling promotes the turnover of the axon survival factor SCG10 (*Shin et al., 2012*), and so hypothesized that MAPK signaling also promotes the turnover of the major axon survival factor NMNAT2. Here we show that inhibition of MAPK signaling slows the turnover of both NMNAT2 and SCG10, leading to increased levels of both survival factors. Moreover, the axon protection afforded by inhibiting MAPK signaling requires elevated levels of NMNAT2. This revises our understanding of how MAPK signaling promotes axon degeneration, placing MAPK signaling upstream of axon survival factor turnover and SARM1 activation. Our new findings unify major axon survival and degeneration proteins into a single, linear pathway.

## Results

### MKK4/7 are necessary for $NAD^+$ loss and axon degeneration after axotomy but not in response to constitutively active SARM1

A linear MAPK signaling pathway is required for axon degeneration. This pathway includes the MAPKKKs DLK, MLK2, and MEKK4, the MAPKKs MKK4 and MKK7 (with MKK4 dominant), and the MAPKs JNK1-3. Within this pathway, the MAPKKs MKK4 and MKK7 represent a targetable bottleneck in which loss of these kinases abrogates the MAPK signaling necessary for axon degeneration

(*Yang et al., 2015*). Lentiviral transduction of previously validated shRNAs targeting the MAPKKs MKK4 and MKK7 into mouse dorsal root ganglia neurons effectively down regulates the levels of MKK4 and MKK7 protein and confers robust axon protection after axotomy when used in tandem (*Figure 3—figure supplement 1*). Thus, by simultaneously knocking down MKK4 and MKK7 (hereafter abbreviated as MKK4/7 or sh4/7), we can interrogate the mechanism by which the MAPK pathway promotes axon degeneration.

Axon injury triggers a number of SARM1-dependent local molecular changes within the axon, including $NAD^+$ depletion and MAPK activation (*Gerdts et al., 2015*; *Yang et al., 2015*). We hypothesized that there could be a linear relationship between $NAD^+$ depletion and MAPK activation such that MAPK signaling is required for $NAD^+$ depletion or, conversely, that $NAD^+$ depletion is the trigger for MAPK activation. To investigate the relationship between MAPK signaling and $NAD^+$ depletion, we assessed whether MAPK signaling is necessary for $NAD^+$ depletion within axons after axotomy. At two and four hours after axotomy, a time when axons remain morphologically intact, axonal $NAD^+$ has decreased to 71% and 7% of baseline levels and is undetectable at six hours after axotomy in neurons expressing a control shRNA that targets luciferase. Strikingly, $NAD^+$ is maintained at 45% of baseline levels six hours after axotomy in the absence of MKK4/7 (*Figure 1A*; p≤0.001), revealing that MAPK signaling is upstream of axotomy-induced $NAD^+$ depletion. ATP depletion is another consequence of axon injury that requires MAPK signaling (*Yang et al., 2015*). ATP levels are also maintained six hours after axotomy when MKK4/7 are knocked down (*Figure 1B*; p≤0.05). These data indicate that MKK4/7 are important for $NAD^+$ and ATP depletion after axotomy.

We previously demonstrated that SARM1 is required for $NAD^+$ and ATP depletion after axotomy (*Gerdts et al., 2015*). Hence, the finding that MKK4/7 are important for axotomy-induced $NAD^+$ and ATP depletion is consistent with MAPK signaling either (a) acting downstream of SARM1 as an intermediate between SARM1 activation and $NAD^+$ and ATP loss, or (b) functioning upstream of SARM1 to promote injury-dependent SARM1 activation. To distinguish between these possibilities, we assessed whether MKK4/7 activity is necessary for $NAD^+$ depletion induced by a constitutively active form of SARM1 that bypasses the need for injury-dependent SARM1 activation. To generate an activated form of SARM1, we used a pharmacologically controlled system in which the SARM1-TIR domain fused to FKBP is homodimerized upon application of the small molecule AP20187 (*Gerdts et al., 2015*; *Yang et al., 2015*). Dimerization of the SARM1-TIR domains induces loss of $NAD^+$ within two hours, similar to the two hour window in which most $NAD^+$ is lost following axotomy (*Figure 1A*). Surprisingly, knockdown of MKK4/7 does not change the rate of $NAD^+$ depletion after SARM1-TIR dimerization (*Figure 1C*). ATP is also depleted, albeit more slowly than $NAD^+$, when SARM1-TIR domains are dimerized. As with $NAD^+$, inhibition of MKK4/7 does not change the rate of ATP depletion induced by SARM1-TIR dimerization (*Figure 1D*). We find similar results when analyzing axonal $NAD^+$ following SARM1-TIR dimerization (SARM1-TIR dimerization, % $NAD^+$ depletion at 2 hr: control 54.9% ± 4.2% vs shMKK4/7 54.9% ± 4.0%, ns, p=0.99). Hence, MAPK signaling is not an intermediate between activated SARM1 and $NAD^+$ depletion. Taken together with the finding that MKK4/7 are required for injury-dependent $NAD^+$ and ATP depletion, these results are consistent with the model that MAPK signaling functions upstream of SARM1 to promote injury-dependent SARM1 activation.

These findings made us reassess the model that MAPK signaling functions downstream of SARM1 in the axon degeneration pathway. The strongest functional evidence for a role for MAPKs downstream of SARM1 in the axon destruction program is the finding that depletion of MKK4/7 partially blocks axon degeneration induced by direct dimerization of the SARM1-TIR domains (*Yang et al., 2015*). We reassessed this finding by simultaneously testing whether MAPKs are required for axon degeneration induced by either axotomy or dimerized SARM1-TIR. In control cultures both axotomy and dimerized SARM1-TIR induce robust axon degeneration. In contrast, depletion of MKK4/7 prevents axon degeneration for at least 24 hr after axotomy; however, in parallel experiments performed in the same dish, depletion of MKK4/7 fails to block axon degeneration induced by dimerization of the SARM1-TIR domains (*Figure 2A–B*). JNK inhibition also fails to block axon degeneration induced by dimerization of the SARM1-TIR domains (*Figure 2C*). These results are consistent with the findings in *Figure 1* demonstrating that MAPK signaling is required for $NAD^+$ depletion following injury but not direct dimerization of SARM1-TIR. While we cannot replicate the finding that the MAPK pathway is required for SARM1-TIR induced axon degeneration, we do

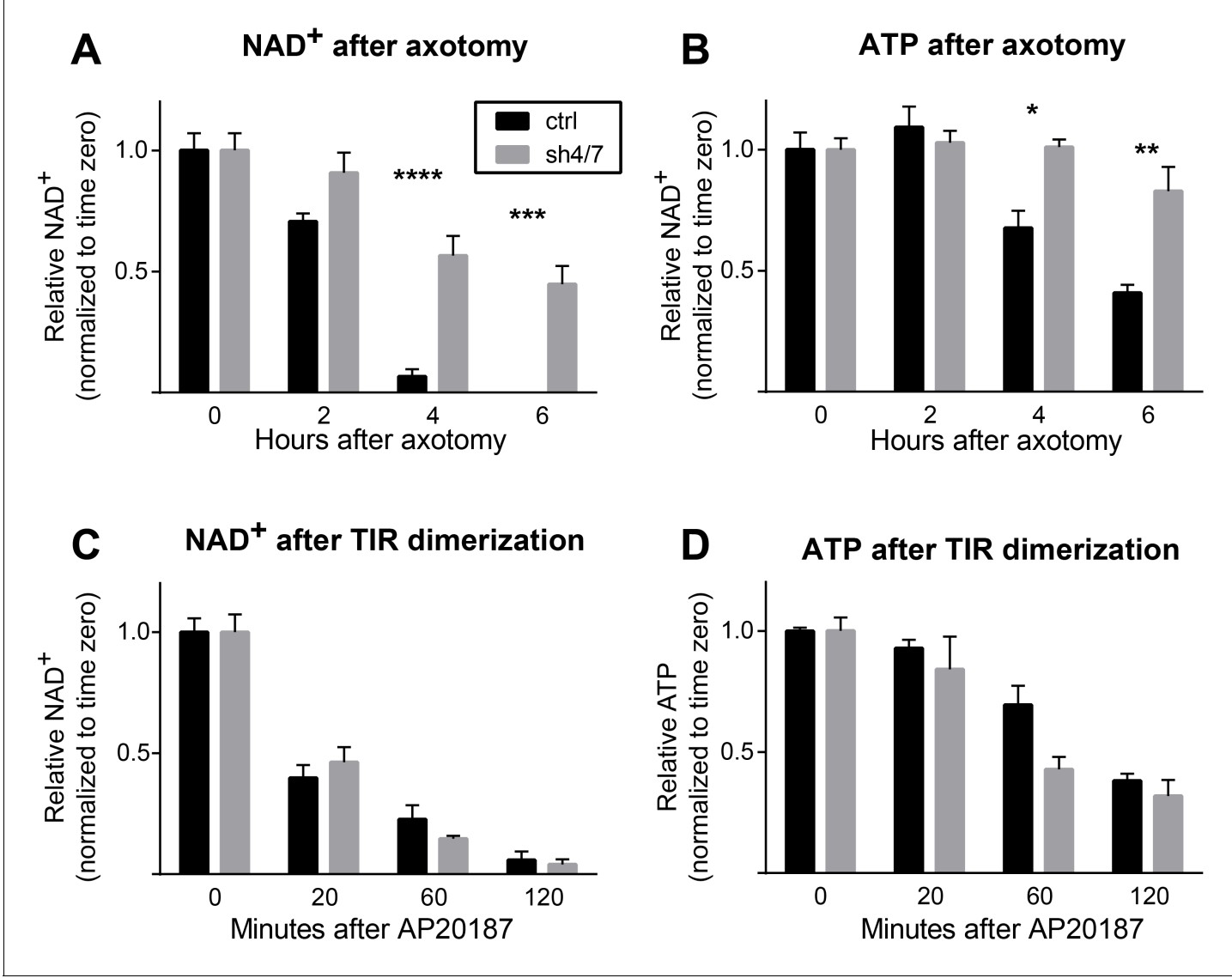

**Figure 1.** MKK4/7 are necessary for NAD$^+$ and ATP loss after axotomy but not in response to constitutively active SARM1. MKK4/7 are necessary for NAD$^+$ and ATP depletion in axons following axotomy but not following direct SARM1 activation. Axonal NAD$^+$ (**A**) and ATP (**B**) levels were assayed at indicated timepoints after axotomy with lentiviral knockdown of MKK4 and MKK7 (sh4/7, grey bars) or Luciferase (ctrl, black bars) control. There is no change in the rate of NAD$^+$ (**C**) or ATP (**D**) depletion after direct activation of SARM1 via dimerization of the SARM1-TIR domains in the absence of injury when MKK4/7 are depleted compared to controls. The compound AP20187 was used to induce SARM1-TIR dimerization. Values are presented as mean ± SEM. All measurements were normalized to time zero. N = 6; p values: *$\leq$ 0.05, **$\leq$ 0.01, ***$\leq$ 0.001, ****$\leq$ 0.0001 by ANOVA.

observe activation of MKK4 following SARM1-TIR dimerization (*Figure 2—figure supplement 1*) as reported by Yang *et al*. Hence, we agree that activated SARM1 can induce the MAPK stress pathway, but we find no functional evidence that this a significant modulator of NAD$^+$ depletion or axon loss. While we cannot explain why our findings differ from those of Yang et al., we suggest that epistasis experiments performed with engineered constructs such as the dimerized SARM1-TIR should be interpreted with care. As such, we chose to investigate the mechanism by which MAPK signaling promotes axotomy-induced axon degeneration. As is shown below, we identify a mechanism by which MAPK signaling acts upstream of endogenous SARM1 to inhibit axon degeneration, and this mechanism is fully consistent with the findings presented here with the dimerized SARM1-TIR.

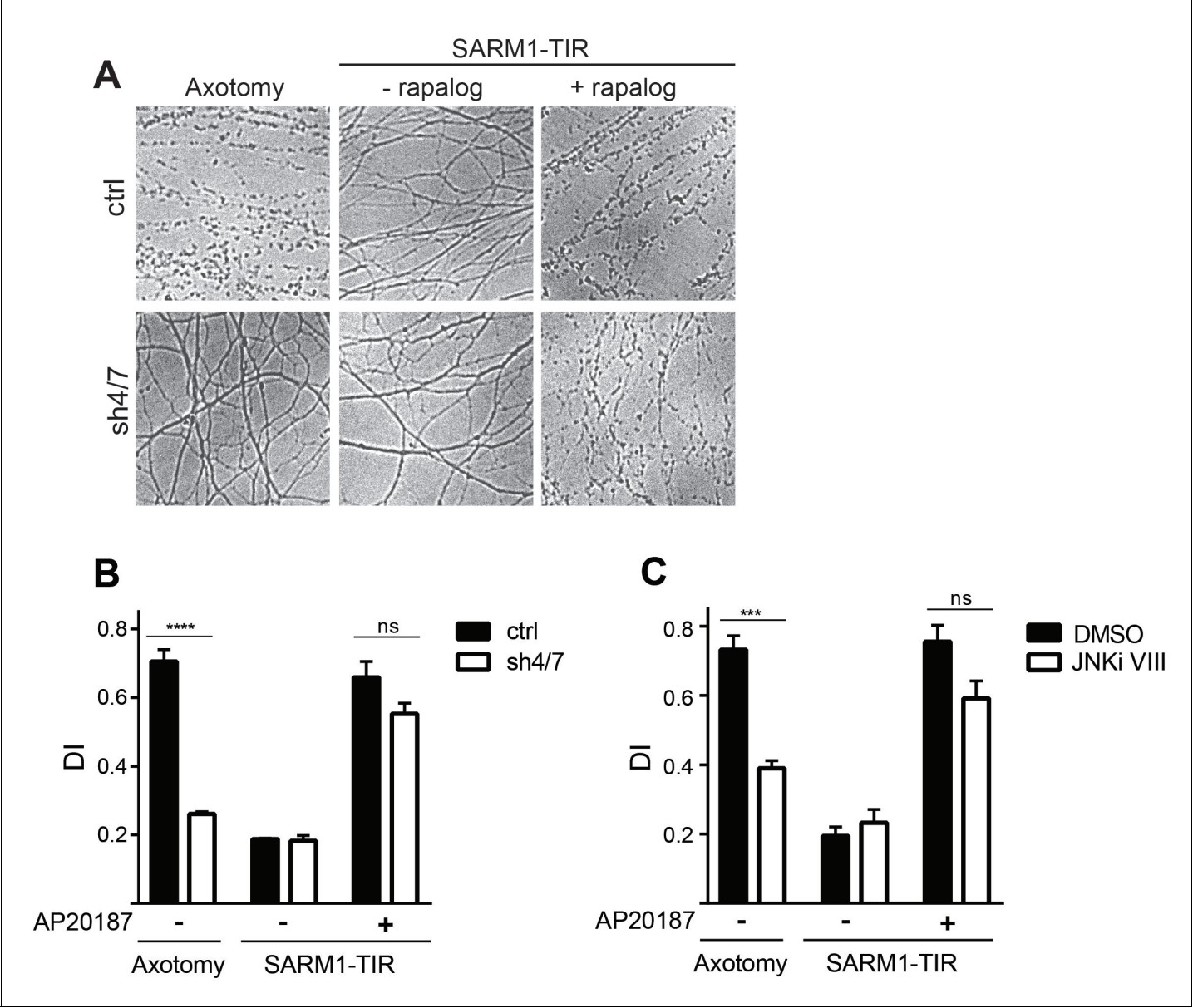

**Figure 2.** MAPKs are required for injury-induced but not SARM1-TIR-induced axon degeneration. (**A**) Depletion of MKK4/7 (sh4/7) protects axons for 24 hours after axotomy but not when axon degeneration is induced by dimerization of the SARM1-TIR domains with addition of the ligand AP20187 (n = 4). Knockdown of MKK4/7 is compared to controls (ctrl) expressing an shRNA targeting luciferase (**B**) Quantification of degeneration index (DI). (**C**) Treatment with JNK inhibitor VIII delays axon degeneration 24 hours after axotomy but not when axon degeneration is induced by dimerization of the SARM1-TIR domains with addition of the ligand AP20187 as compared to DMSO vehicle control (n = 3). The data for each histogram are derived from neurons that were cultured in the same dish with treatments (i.e. axotomy or TIR-dimerization) performed in parallel. p value: ***≤ 0.001 and ****≤ 0.0001 by ANOVA; non-significant (ns). Values are presented as mean ± SEM. See also *Figure 2—figure supplement 1*.

The following figure supplement is available for figure 2:

**Figure supplement 1.** Dimerization of the SARM1-TIR domains with the ligand AP20187 results in MKK4 phosphorylation on Ser257/Thr261.

## MKK4/7 regulates the levels of axon survival factors NMNAT2 and SCG10

The mechanism of injury-dependent SARM1 activation is unknown, however one prominent model is that the axonal survival factor NMNAT2 delivered from the cell body inhibits SARM1 activation in

healthy axons (*Gilley et al., 2015*). We previously demonstrated that JNK promotes the turnover of a second axonal survival factor, the microtubule regulating protein SCG10, whose overexpression can also delay axon degeneration, albeit much less potently than NMNAT2 (*Shin et al., 2012*). This lead us to test the hypothesis that MAPK signaling might have a broader role in controlling the stability of axonal maintenance factors. We tested if MAPKs promote turnover of survival factors by assessing whether depletion of MKK4/7 leads to an increase in the levels of NMNAT2 and SCG10 protein. We first compared levels of NMNAT2 and SCG10 in the presence or absence of MKK4/7 under basal conditions. We find that levels of endogenous NMNAT2 and SCG10 are elevated in neurons upon MKK4/7 knockdown (*Figure 3A* and quantified in 3F; NMNAT2 $3.2 \pm 0.5$ fold increase, SCG10 $5.4 \pm 1$ fold increase). We observe a similar increase in the levels of NMNAT2 and SCG10 when we knockdown MKK4 with a second shRNA, and when we target MKK4 using the Cas9/CRISPR system (*Figure 3—figure supplement 1B*). Axon degeneration is regulated locally; therefore, we asked if MKK4/7 regulates axon survival factors within axons. Levels of endogenous NMNAT2 and SCG10 are elevated within axon-only lysate after depletion of MKK4/7 (*Figure 3B*; Nmnat2 $3.5 \pm 0.8$ fold increase, SCG10 $4.3 \pm 1.3$ fold increase). Similar results were observed following treatment with the JNK inhibitor VIII, where levels of NMNAT2 and SCG10 are boosted within two hours of JNK inhibition (*Figure 3C*). This finding indicates that MKK4/7 function through the canonical JNK pathway to deplete NMNAT2 and SCG10.

We analyzed SCG10 and NMNAT2 expression by immunocytochemistry as an independent test of whether these survival factors are elevated within axons. While there is an excellent antibody for SCG10 (*Shin et al., 2012*), our antibody for endogenous NMNAT2 works for Westerns but not immunocytochemistry. Hence we expressed myc-tagged NMNAT2 (NMNAT2-myc) in DRG neurons from the ubiquitin C promoter. Upon MKK4/7 knockdown, both NMNAT2-myc and SCG10 are elevated within axons (*Figure 3D*; $p \leq 0.001$, n = 7–9). Moreover, the upregulation of NMNAT2-myc expressed from an exogenous promoter indicates that MKK4/7 regulates NMNAT2 post-transcriptionally, consistent with the hypothesis that they regulate protein turnover (*Figure 3—figure supplement 1D*).

Having demonstrated that MAPKs regulate levels of NMNAT2 in vitro in mammalian neurons, we next assessed whether this regulation is evolutionarily conserved and occurs in vivo. The JNK ortholog Basket (Bsk) promotes axon degeneration of motoneurons in *Drosophila* larvae—expression of a dominant negative JNK transgene or depletion of JNK using RNAi delays axon degeneration after nerve pinch injury (*Figure 4—figure supplement 1*, *Xiong and Collins, 2012*). Thus, we asked if JNK can also regulate levels of the only NMNAT isoform in *Drosophila*, dNmnat. We expressed HA-dNmnat in motoneurons using OK6-Gal4 and assessed levels of tagged dNmnat by immunostaining. We find that expression of HA-dNmnat is elevated within larval nerves when MAPK activity is inhibited (*Figure 4*; JNK dominant negative $2.8 \pm 0.2$ fold higher; JNK RNAi $2.2 \pm 0.2$ fold higher than controls). These results demonstrate that MAPKs regulate NMNAT in vivo, and that this MAPK-dependent regulation is evolutionarily conserved.

MKK4/7 function upstream of SARM1 for NAD$^+$ depletion (*Figure 1*), however there are strong data that SARM1 and its orthologs can function upstream of MAPK signals (*Figure 2—figure supplement 1*; *Blum et al., 2012*; *Chen et al., 2011*; *Chuang and Bargmann, 2005*; *Kurz et al., 2007*; *Yang et al., 2015*). Hence, we tested whether the MKK4/7-dependent regulation of axonal survival factors occurs downstream of SARM1. If MKK4/7 function downstream of SARM1, then we predict that loss of SARM1 should phenocopy the loss of MKK4/7 and lead to an increase in the levels of NMNAT2 and SCG10. However, levels of NMNAT2 and SCG10 are comparable in wild type and SARM1 knockout neurons (*Figure 3E–F*; not significant, n = 3; *Gilley et al., 2015*). If SARM1 were upstream of MKK4/7, then we would further predict that MKK4/7 regulation of survival factors would require SARM1. To test this prediction, we knocked down MKK4/7 in both wild type and SARM1 knockout neurons and measured the levels of NMNAT2 and SCG10. We find that MKK4/7 regulate these axon survival factors even in the absence of SARM1, as levels of NMNAT2 and SCG10 are elevated upon knockdown of MKK4/7 in both wild type and SARM1 knockout neurons (*Figure 3E–F*; in SARM1 knockout with shMKK4/7, NMNAT2 $3.9 \pm 0.7$ fold increase, SCG10 $6.6 \pm 1.4$ fold increase compared to shRNA controls; not significant comparing genotypes). Thus, MAPK regulation of axon survival factors is independent of SARM1. Since elevated levels of NMNAT2 inhibits SARM1-dependent axon degeneration, these findings are consistent with the model that MAPK signaling functions upstream of SARM1.

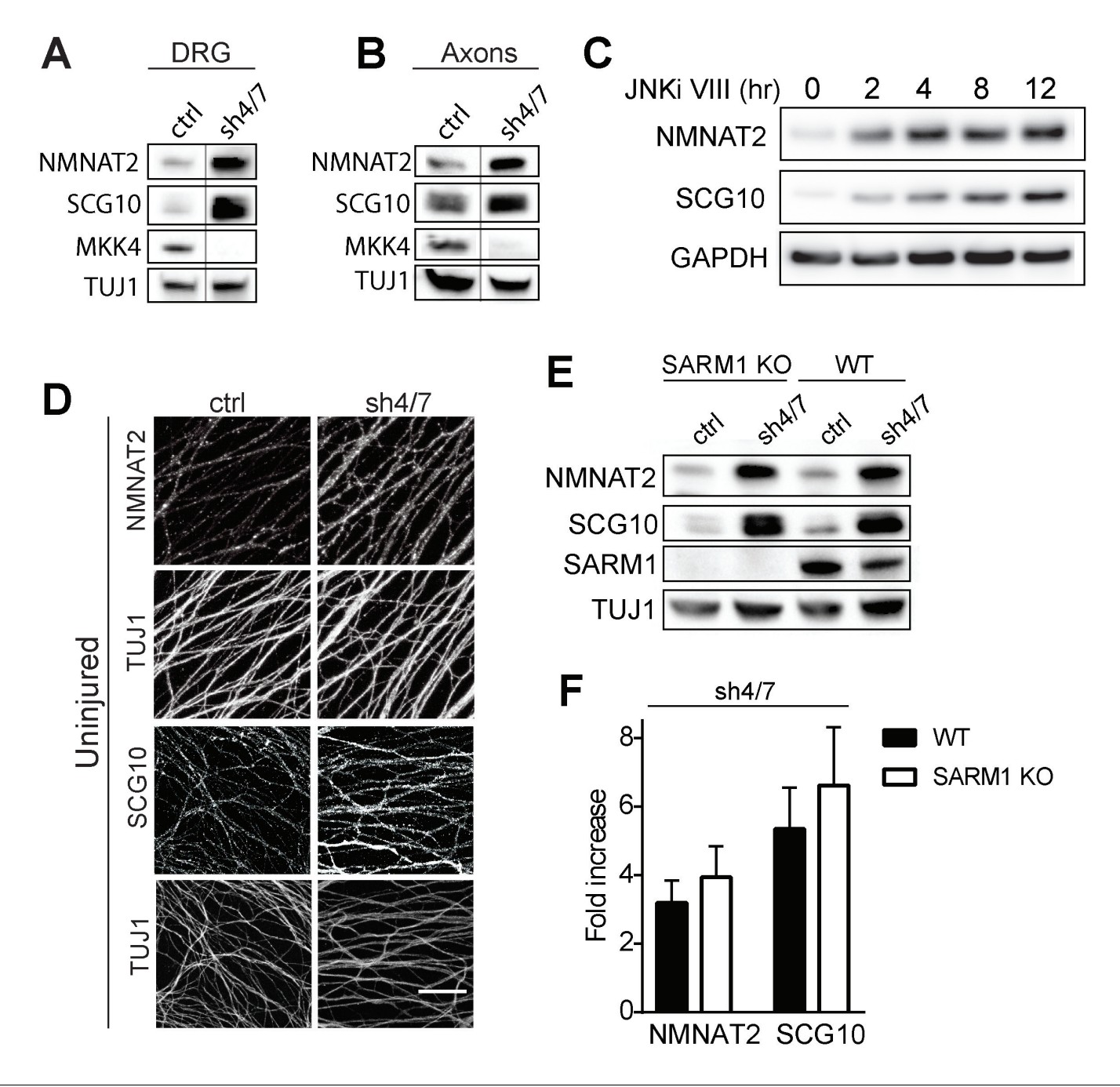

**Figure 3.** MKK4/7 regulates the levels of axon survival factors NMNAT2 and SCG10. Loss of MKK4/7 (sh4/7) increases levels of endogenous NMNAT2 and SCG10 within cultured DRG neurons (**A**) and axons (**B**) as observed via western blot as compared to shLuciferase (ctrl) controls (n = 3). Vertical line in westerns blots denotes where lanes were removed; however, samples were run on the same gel and images are from the same exposure. We confirm that MKK4/7 functions through the canonical JNK pathway, as timecourse treatment with JNK inhibitor (JNKi) VIII (10 uM, 0–12 hr treatment; n = 4) increases endogenous NMNAT2 and SCG10 within neurons within two hours. (**C**) MKK4/7 regulate levels of exogenously-expressed NMNAT2-myc and endogenous SCG10 (**D**) as observed by immunostaining within uninjured DRG axons, indicating the modulation is post-transcriptional. The anti-Tuj1 antibody stains tubulin and labels all axons. Scale bar: 25 μm, n = 4 (**E**) Western blot from embryonic DRG neurons reveals that knockdown of MKK4/7 leads to an increase in the levels of endogenous NMNAT2 and SCG10 in both wildtype (WT) and SARM1 knockout (SARM1 KO) neurons. (**F**) Quantification of endogenous NMNAT2 and SCG10 in WT and SARM1 knockout neurons with depletion of MKK4/7 by shRNA. Data were normalized to protein levels with shLuciferase controls (not significant, n = 3). Also refer to *Figure 3—figure supplement 1*.

*Figure 3 continued on next page*

*Figure 3 continued*

The following figure supplement is available for figure 3:

**Figure supplement 1.** Depletion of MAPKs elevates survival factors and protects axons.

## MKK4/7 promote turnover of axon survival factors

Survival factors function locally within the axon after injury. Thus, we asked whether inhibition of MKK4/7 maintains elevated levels of NMNAT2 and SCG10 within the axon in the first hours after injury, when the axon commits to degenerate. We axotomized cultured DRG neurons and stained for myc-tagged NMNAT2 and endogenous SCG10. By four hours after axotomy, NMNAT2 is depleted and endogenous SCG10 is undetectable in distal axons from control neurons. However, four hours after axotomy NMNAT2 and SCG10 are maintained within distal axons from MKK4/7 knockdown neurons (*Figure 5A*). Indeed, axonal levels of NMNAT2 and SCG10 are comparable at four hours post-axotomy from MKK4/7 knockdown neurons to the levels in uninjured axons from wild type neurons (for NMNAT2-myc, fluorescence intensity four hours after injury for shMKK4/7 is $1.2 \pm 0.2$ fold ($p>0.05$) of the levels measured in uninjured control axons; for SCG10, fluorescence intensity four hours after injury for shMKK4/7 is $1.1 \pm 0.2$ fold ($p>0.05$) of the levels measured in uninjured control axons ($p>0.05$), n = 7–9). The four hour time-point is significant, as this is approximately the time at which axons irreversibly commit to degenerate after injury in vitro (*Gerdts et al., 2016*). Hence, knockdown of MKK4/7 maintains axon survival factors at the time and place they are needed to inhibit the axon degeneration program.

Both NMNAT2 and SCG10 are labile proteins that are rapidly turned over in axons. We previously demonstrated that JNK inhibition slows the turnover rate of SCG10 (*Shin et al., 2012*). Hence, we postulated that MAPK signaling limits the levels of axon survival factors by promoting their turnover. To test this hypothesis, we assessed their half-life by treating cultured DRG neurons with the translation inhibitor cycloheximide to block new protein synthesis and measured the levels of endogenous NMNAT2 and SCG10 within axons at various times after cycloheximide treatment (*Figure 5C–E*). Consistent with previous reports (*Gilley and Coleman, 2010*; *Shin et al., 2012*), the normal turnover of NMNAT2 and SCG10 is rapid, with half-lives of approximately 1.1 and 1.5 hr, respectively. However, with MKK4/7 knockdown the turnover of these survival factors is slowed, with half-lives for NMNAT2 and SCG10 of approximately 3.4 and 2.7 hr, respectively. Thus, MAPKs likely influence the turnover of NMNAT2 and SCG10.

As levels of NMNAT2 and SCG10 are elevated within the axon upon MAPK depletion, it is possible that turnover is delayed due to saturation of degradation machinery. To test this hypothesis we performed two additional experiments. First, we pre-treated DRG neurons with JNK inhibitor VIII for only 30 min, which does not allow for enough time for substantial buildup of protein. We then treated with cycloheximide and measured endogenous NMNAT2 and SCG10 turnover in axons. We find that even acute inhibition of JNK signaling is sufficient to delay the degradation of both survival factors, consistent with the model that MAPK signaling regulates protein turnover (*Figure 5—figure supplement 1*). Second, we tested directly whether high levels of axonal NMNAT2 and SCG10 would hinder the degradation process and slow turnover. For this, we compared the turnover rate within axons of endogenous NMNAT2 and SCG10 to the turnover rate in cultures overexpressing high levels of NMNAT2-myc and wildtype SCG10. We observe no change in the degradation rate of survival factors when levels are increased over 5-fold higher than endogenous levels, which is even more than the increase in protein levels that we observe with depletion of MKK4/7 (*Figure 5—figure supplement 2*). Hence, the slowed turnover is not due to the higher levels of these proteins. Finally, there is no increase in transcript levels of either NMNAT2 or SCG10 after depletion of MKK4/7 as assessed by rt-PCR (*Figure 5B*). These findings are all consistent with the model that MAPK signaling promotes the turnover of NMNAT2 and SCG10. While turnover is slowed, NMNAT2 and SCG10 are still lost in the absence of MKK4/7, albeit at a slower rate, indicating that there are likely MAPK-independent mechanisms promoting their degradation.

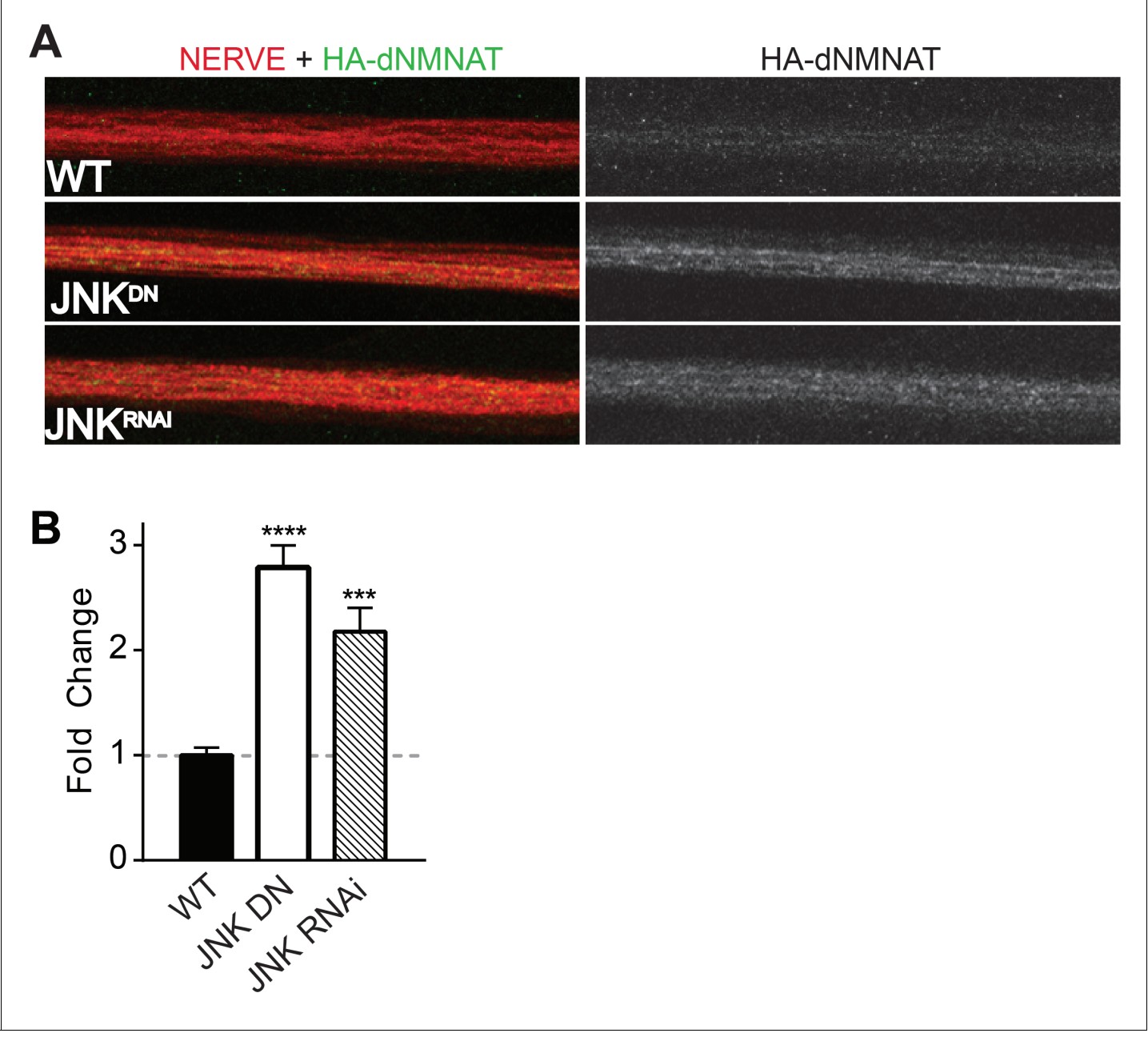

**Figure 4.** MAPKs modulate dNMNAT in vivo in *Drosophila* larvae. Immunostaining in third instar *Drosophila* larvae of HA-dNMNAT (green) expressed in nerves from the motorneuron driver OK6-Gal4, with HRP labeling (red) to counterstain the nerve. (**A**) Expression of JNK dominant negative (DN) or depletion of JNK with RNAi (BL57035) increases levels of HA-dNMNAT in the nerve in vivo. (**B**) Quantification of fluorescence intensity of HA-dNMNAT normalized to wildtype (WT) controls. N = 8–10 animals. p values: ***≤ 0.001, ****≤ 0.0001 by ANOVA. Also refer to *Figure 4—figure supplement 1*.

The following figure supplement is available for figure 4:

**Figure supplement 1.** Depletion of JNK with RNAi or expression of JNK dominant negative (DN) protects *Drosophila* larval axons 24 hr after pinch injury.

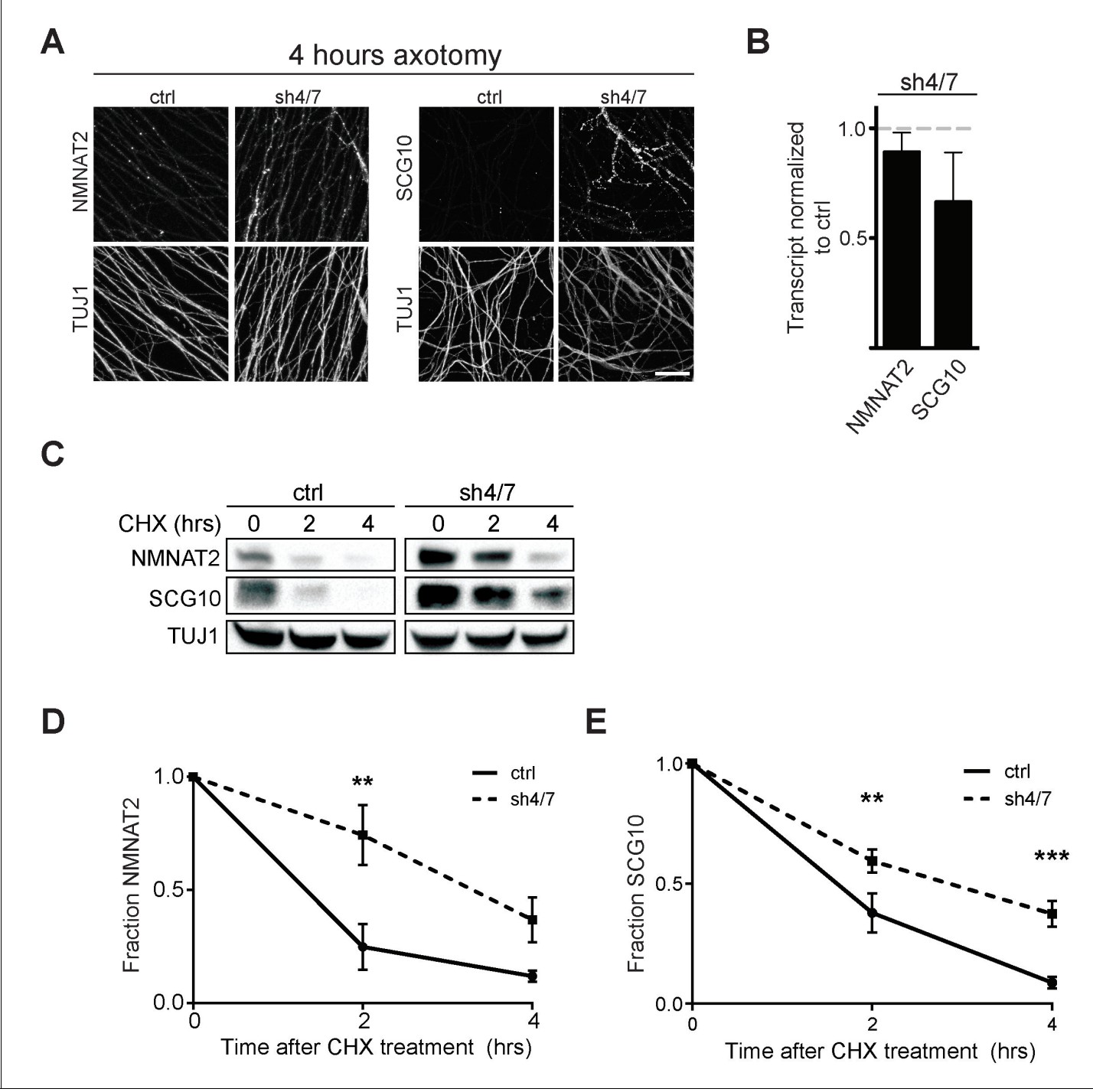

**Figure 5.** MKK4/7 promote turnover of axon survival factors. (**A**) Immunostaining for NMNAT2-myc and SCG10 in distal axons four hours after axotomy. Fluorescence signal is diminished four hours after cut in control axons; however, with depletion of MKK4/7 (sh4/7) levels of NMNAT2-myc and SCG10 are maintained after injury compared to shLuciferase (ctrl) control (p≤0.01 for NMNAT2-myc and p≤0.001 for SCG10, comparing fluorescence between conditions at four hours, n = 4). Scale bar: 25 µm (**B**) rtPCR reveals no significant change in transcript levels for NMNAT2 or SCG10 after depletion of MKK4/7 when normalized to shLuciferase controls. N = 3, p=0.29. (**C**) Western blot analysis from axon-only lysate from DRG cultures treated with cycloheximide (CHX) for the indicated timepoints to block protein synthesis. Quantification from CHX timecourse experiments for fraction of NMNAT2 (**D**) and SCG10 (**E**) as compared to time zero. The turnover rate of both NMNAT2 and SCG10 is slowed upon MKK4/7 knockdown. N = 4; p values: **≤ 0.01, ***≤ 0.001. Also refer to *Figure 5—figure supplements 1* and *2*.

The following figure supplements are available for figure 5:

*Figure 5 continued on next page*

*Figure 5 continued*

**Figure supplement 1.** Pre-treatment with JNKi VIII (30 min) delays degradation of NMNAT2 and SCG10 after cycloheximide (CHX).

**Figure supplement 2.** Elevation of NMNAT2 and SCG10 does not delay turnover of survival factors within axons.

## MAPK signaling promotes axonal degeneration via regulation of survival factors

Inhibition of MAPK signaling via knockdown of MKK4/7 delays axonal degeneration following axotomy (see *Figure 3—figure supplement 1C*, *Yang et al., 2015*) and increases the levels of NMNAT2 (*Figure 3*). Since increased levels of NMNAT2 are sufficient to delay axon degeneration following injury, we hypothesized that the axonal protection afforded by MKK4/7 knockdown is due to the elevation of NMNAT2 levels. To test this model, we simultaneously depleted MKK4/7 with shRNAs and used guide RNAs (gRNAs) to knockout NMNAT2 in cultured Cas9 knockin DRG neurons and assessed the rate of axon degeneration following axotomy. Knockout of NMNAT2 using gRNAs fully blocks the increase in NMNAT2 protein levels induced by MKK4/7 knockdown (*Figure 6—figure supplement 1*), and also abrogates the axonal protection resulting from MKK4/7 knockdown (*Figure 6A–B* and *Figure 6—figure supplement 1*). Following axotomy, axons from control neurons degenerate after 6 hr, yet 24 hr after axotomy axons from MKK4/7 knockdown neurons remain intact with addition of control scrambled gRNA (*Figure 6A–B* and *Figure 6—figure supplement 1*). However, this protection is lost when NMNAT2 is simultaneously knocked out with gRNAs (*Figure 6A–B* and *Figure 6—figure supplement 1*). Indeed, with the simultaneous knockdown of MKK4/7 and NMNAT2 axons degenerate at 6 hr after axotomy, the same time point at which wildtype axons begin to degenerate. Hence, NMNAT2 is required for the axonal protection afforded by MKK4/7 knockdown, indicating that the protective effect of inhibiting MAPK signaling is conferred by elevated levels of NMNAT2. NMNAT2 knockdown does not block the protection afforded by deletion of SARM1 (*Figure 6A–B*), consistent with previous findings showing that NMNAT2 functions upstream of SARM1 to inhibit axonal degeneration (*Gilley et al., 2015*).

These findings demonstrate that NMNAT2 is absolutely essential for the protection mediated by loss of MAPK signaling, raising the question of whether or not SCG10 also participates in this process. While overexpression of NMNAT2 and SCG10 each confer axon protection after injury (*Gilley and Coleman, 2010*; *Shin et al., 2012*), inhibition of MAPK signaling elevates the levels of these proteins in tandem. To test of whether SCG10 could enhance NMNAT2-induced axon protection, we simultaneously overexpressed NMNAT2 and SCG10 in order to mimic this coordinate up regulation. We overexpressed SCG10-AA (S62/73A), a more stable form of the protein in which JNK phosphorylation sites are mutated (*Shin et al., 2012*), and NMNAT2-myc singly or in combination in otherwise wild type cultured DRG neurons. Overexpression of SCG10-AA delays axon degeneration after axotomy by about 3 hr compared to control, consistent with the very modest axon protection previously described (*Shin et al., 2012*). Overexpression of NMNAT2-myc protects axons for 36 hr after axotomy. Surprisingly, co-expression of SCG10-AA and NMNAT2-myc results in robust axon protection for at least 72 hr (*Figure 6C–D*). The extent of protection is comparable to that achieved via overexpression of cytoplasmic NMNAT1, the most axo-protective gain-of-function manipulation described to-date (*Babetto et al., 2010*; *Sasaki et al., 2009a*). Thus, elevation of NMNAT2 and SCG10 confers robust protection, indicating that these factors either function synergistically or contribute to parallel pathways to prevent axon degeneration. These results are consistent with the model that MAPK signaling promotes axonal degeneration by coordinately down regulating axon survival factors (*Figure 7*).

## Discussion

A balance between axonal survival and axonal degeneration factors determines the choice between axon maintenance and axon loss. MAPK signaling via JNK is an essential component of the pro-degenerative program (*Miller et al., 2009*; *Yang et al., 2015*), but the mechanism of action of MAPK signaling was not known. Here we define this mechanism, demonstrating that MAPK signaling

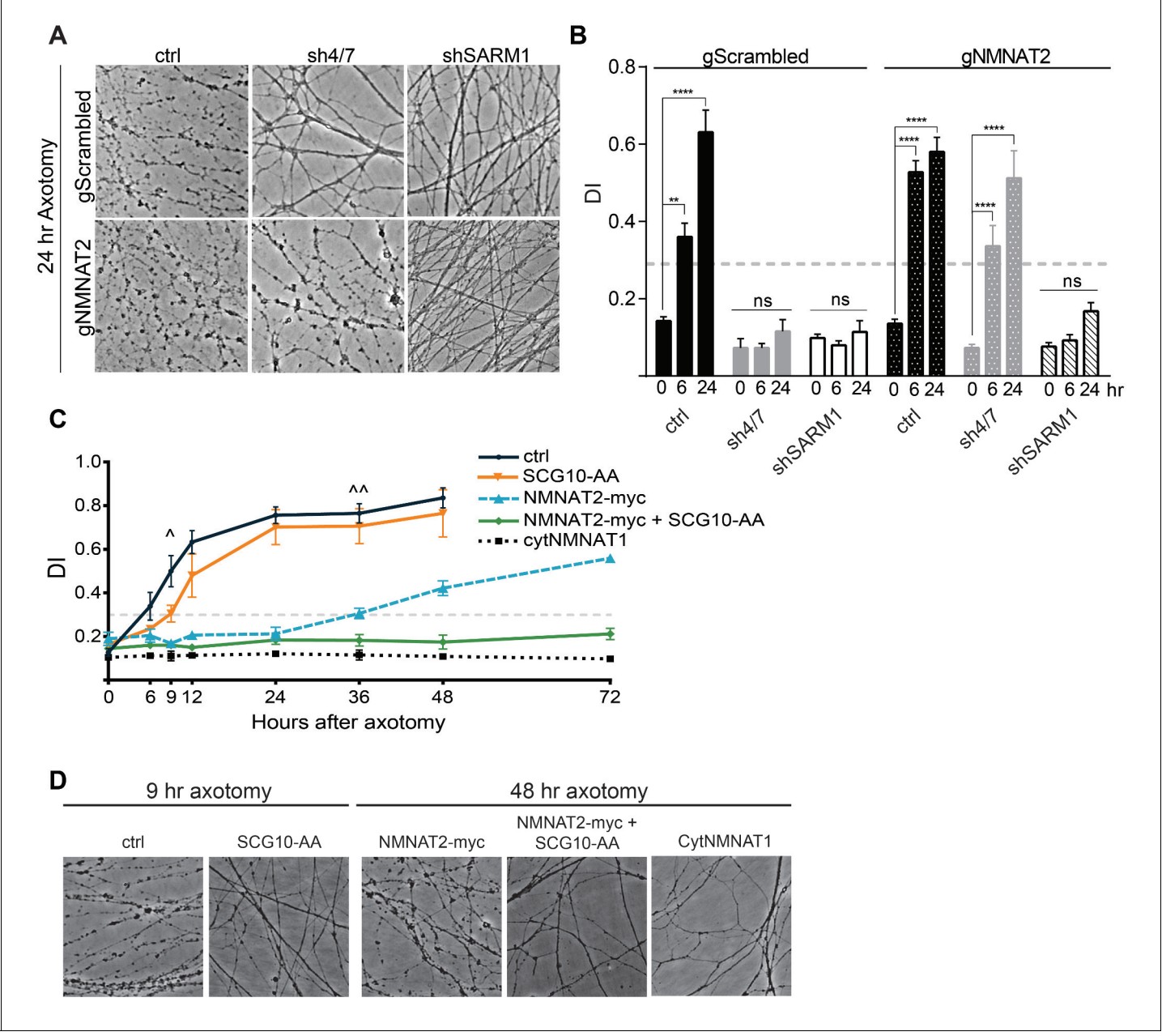

**Figure 6.** MAPK signaling promotes axonal degeneration via regulation of survival factors. MKK4/7 requires NMNAT2 to protect axons. Cas9 knock-in DRGs were cultured and treated with scrambled (gScrambled) or NMNAT2 (gNMNAT2) guide RNAs. (A) Knockdown of MKK4/7 (sh4/7) protects axons at 24 hr after axotomy; however, knocking out NMNAT2 using guide RNAs suppresses this protection (p≤0.0001). In contrast, axons from SARM1 $^{-/-}$ DRG neurons do not require NMNAT2 to maintain integrity. (B) Quantification of degeneration index (DI) from images shown in A and in *Figure 6— figure supplement 1* plus additional replicates. NMNAT2 guide RNAs revert protection from shMKK4/7 at 6 hr. (C) The protection conferred by overexpression of Scg10-AA (stable, with JNK sites mutated) and NMNAT2-myc is synergistic. Expression of SCG10-AA protects axons for 9 hr (^), while overexpression of NMNAT2-myc delays degeneration for 36 hr (^^). When expressed together, SCG10-AA and NMNAT2-myc protect axons for 72 hr, comparable to protection due to expression of cytoplasmically-localized NMNAT1 (cytNMNAT1). At 9 hr, all conditions are statistically different from non-recombinant vector controls (ctrl; p≤0.01) and axons are protected. After 48 hr, NMNAT2-myc is statistically different from cytNMNAT1 and NMNAT2-myc + SCG10-AA (p≤0.01); however, cytNMNAT1 and NMNAT2-myc + SCG10-AA are essentially equivalent. The horizontal dotted lines indicate the DI value above which axons have begun to degenerate. (D) Representative images from figure C at 9 hr or 48 hr after axotomy, as noted. N = 3–4. Also refer to *Figure 6—figure supplement 1*.

The following figure supplement is available for figure 6:

**Figure supplement 1.** MKK4/7 requires NMNAT2 to protect axons.

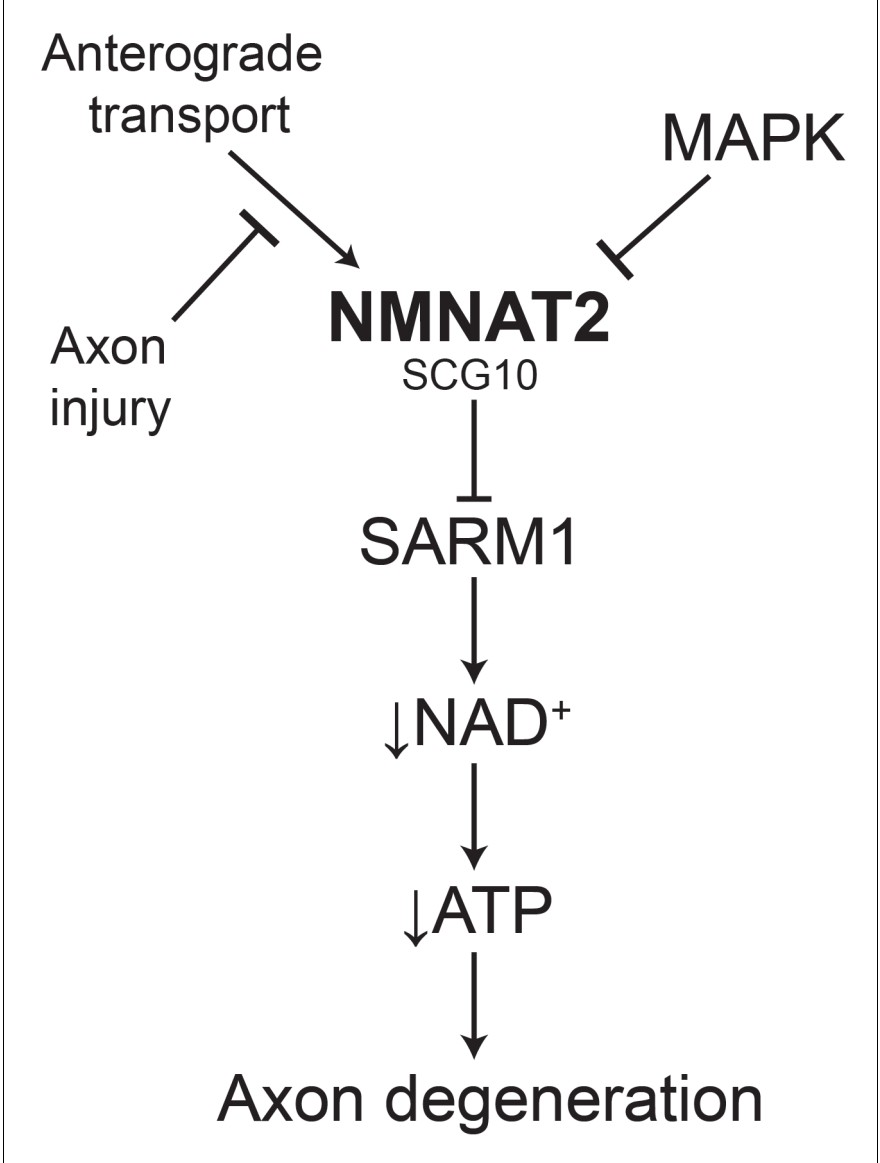

**Figure 7.** Model of the axonal degeneration program. In this model, MAPK signaling limits the levels of NMNAT2 and SCG10. NMNAT2 blocks activation of SARM1, and SCG10 may facilitate NMNAT2 function. Upon axon injury, delivery of the labile survival factors via anterograde transport is blocked. The continued degradation of survival factors allows levels of survival factors to drop below a critical threshold, thereby activating SARM1. Activated SARM1 rapidly depletes NAD$^+$, leading to depletion of ATP and induction of a metabolic crisis that drives axon fragmentation. Our studies identify a critical function for MAPK signaling upstream of endogenous SARM1, however forced dimerization of SARM1-TIR can activate JNK, and so there could be additional roles for MAPK signaling downstream of SARM1.

promotes axonal degeneration by enhancing the turnover of axonal survival factors. We present a model of the axonal degeneration program that incorporates many of the major axon survival and degeneration proteins into a linear pathway.

SARM1 is the central executioner of the axonal degeneration program, and so it is important to understand how activated SARM1 triggers axon loss. Others and we have identified two downstream consequences of SARM1 activation. We demonstrated that activated SARM1 induces the rapid degradation of the essential metabolic co-factor NAD$^+$, and that endogenous SARM1 is necessary for NAD$^+$ loss following nerve injury in vivo (*Gerdts et al., 2015*). Yang and colleagues showed

that activated SARM1 induces rapid MAPK activation, and that endogenous SARM1 is necessary for MAPK activation after nerve injury in vivo (*Yang et al., 2015*). Here we tested the hypothesis that MAPK signaling is necessary for SARM1-dependent NAD$^+$ loss. To our surprise, MAPK signaling is necessary for injury-dependent NAD$^+$ depletion, but not for NAD$^+$ loss induced by activated SARM1. MAPK signaling is also necessary for axon degeneration induced by axotomy, but not axon degeneration induced by activated SARM1. These findings support the model that MAPK signaling is required for the injury-induced activation of SARM1, but that once SARM1 is active, then MAPK signaling is dispensable for SARM1-dependent NAD$^+$ depletion and axon loss.

The detailed molecular mechanism of injury-dependent SARM1 activation is unknown, however strong genetic data demonstrate that loss of the axonal survival factor NMNAT2 induces SARM1-dependent axon degeneration (*Gilley et al., 2015*). Here we show that MAPK signaling promotes axonal NMNAT2 protein turnover both in mouse DRG neurons in culture and *Drosophila* motoneurons in vivo. Moreover, this increase in NMNAT2 levels is necessary for the axon protection afforded by knockdown of MKK4 and MKK7. In addition, MAPK signaling promotes the turnover of the only other identified axonal survival factor, SCG10. Taken together, these data fit the model that MAPK signaling promotes axonal degeneration by limiting the levels of axon survival factors.

This model places MAPK signaling upstream of SARM1 activation. However, SARM1 activity can induce MAPK activation (this manuscript; *Chang et al., 2011*; *Hayakawa et al., 2011*; *Inoue et al., 2013*; *Tong et al., 2009*; *Yang et al., 2015*), which would be consistent with an alternative model in which SARM1 activates MAPK signaling which then promotes the turnover of NMNAT2 as part of a positive feedback loop. We tested and excluded this feedback loop model using SARM1 knockout neurons. SARM1 knockout neurons do not have elevated levels of NMNAT2 or SCG10 at steady state, and the regulation of NMNAT2 and SCG10 by MKK4/7 does not require SARM1. The Coleman group reported similar findings in injured axons, demonstrating that NMNAT2 is still degraded after injury in SARM1 knockout neurons (*Gilley et al., 2015*). These data are incompatible with the feedback loop model. Instead, the finding that MKK4/7 regulate the levels of these axon survival factors in the absence of SARM1 is genetic proof that for this activity MAPK signaling does not function downstream of SARM1. However, it does not exclude the possibility that SARM1-dependent activation of MAPK signaling plays a modulatory role that was not discernable in our functional studies.

How might MAPK signaling promote the turnover of NMNAT2 and SCG10? For SCG10, the regulation is, at least in part, direct. SCG10 is phosphorylated by JNK kinases (*Shin et al., 2012*; *Tararuk et al., 2006*), and we showed that mutating these phosphorylation sites slows SCG10 turnover in axons (*Shin et al., 2012*). However, we also demonstrated that inhibiting JNK kinases further slows the turnover of this mutant SCG10, suggesting additional mechanisms of regulation (*Shin et al., 2012*). For NMNAT2 we do not know the mechanism. We have mutated candidate phosphorylation sites, but have not identified sites required for MAPK regulation (unpublished findings). NMNAT2 turnover is regulated by palmitoylation, localization, and the ubiquitin ligase Highwire/PHR1 (*Babetto et al., 2013*; *Milde and Coleman, 2014*; *Milde et al., 2013a*; *Xiong et al., 2012*). Hence MAPK signaling could impact NMNAT2 turnover by modulating these processes, and this will be a topic of future studies.

Our findings in conjunction with previous work in the field support a simple model for the mechanism of axonal degeneration that incorporates many of the major axonal survival and degenerative proteins (*Figure 7*). In this model MAPK signaling limits the levels of NMNAT2, and NMNAT2 in turn inhibits the activation of SARM1. Upon injury, anterograde transport of NMNAT2 is blocked, thereby causing a local reduction in NMNAT2 as a consequence of constitutive protein degradation. When NMNAT2 levels fall below a critical threshold, SARM1 is activated, leading to a precipitous drop in NAD$^+$ and ATP levels, energetic crisis, and axon destruction. In this model, MAPK activity is a rheostat controlling the levels of NMNAT2 and hence the propensity for an axon to degenerate. Such tuning might be particularly important in chronic conditions such as neuropathy in which axonal transport of survival factors may be impaired but not completely blocked, as it is following axotomy.

In this model the core components of the degeneration program form a linear pathway, but additional regulatory molecules will impinge on this mechanism. For example, AKT activity is mildly axoprotective, and inhibits MAPK signaling via direct phosphorylation of MKK4 (*Wakatsuki et al., 2011*; *Yang et al., 2015*). The ubiquitin ligase Highwire/PHR1 targets NMNAT2 for degradation, and loss of this ligase is potently axoprotective (*Babetto et al., 2013*; *Xiong et al., 2012*). The axoprotective mechanism of action of SCG10 is unknown, but we previously demonstrated that maintaining

SCG10 in injured axons enhances axonal transport (*Shin et al., 2012*). We show that co-expression of SCG10 with NMNAT2 greatly enhances axon protection after injury, and so speculate that SCG10 could facilitate the transport of NMNAT2 to its sites of action within the injured axon. To date no proteins have been identified that directly regulate SARM1, but we anticipate that such molecules would potently modulate axon degeneration. We suggest that knowledge of the core axonal degeneration program presented here will provide a framework for understanding the molecular mechanism of newly identified players in this pathway.

## Materials and methods

### Antibodies and chemicals

The following antibodies were used in this study: rabbit anti-$\beta$3 tubulin (TUJ1; 1:5000; Sigma-Aldrich (St. Louis, MO) Cat# T2200 RRID:AB_262133), mouse anti-$\beta$3 tubulin (Tuj1; 1:5000; Biolegend, (San Diego, CA) Cat# 801202 RRID:AB_10063408), rabbit anti-MKK4 (1:1000; Cell Signaling Technologies (CST; Danvers, MA) Cat# 9152 RRID:AB_330905), rabbit anti-MKK7 (1:1000; CST Cat# 4172 RRID: AB_330914), rabbit HSP90 (1:500; CST Cat# 4877S RRID:AB_2233307), rabbit Phospho-SEK1/MKK4 (Ser257/Thr261) (1:1000; CST Cat# 9156S RRID:AB_2297420), mouse anti-Myc-Tag (9B11; 1:5000; CST Cat# 2276 RRID:AB_2314825), rabbit anti-SCG10 (*Shin et al., 2012*), rabbit anti-NMNAT2 (*Mayer et al., 2010*), Cy3 and Cy5-conjugated goat $\alpha$-HRP (1:1000, Jackson ImmunoResearch (West Grove, PA)), Alexa488 $\alpha$-GFP (1:1000, Life Technologies (Carlsbad, CA)), $\alpha$-HA (16B2; 1:500, Covance). JNK inhibitor VIII (#15946, Cayman Chemical Company (Ann Arbor, MI)) was used at 10 $\mu$M, AP20187 (BB Homodimerizer; Clontech (Mountain View, CA) #635060) was used at 50 nM, and cycloheximide (10 ug/mL, Sigma (St. Louis, MO)).

### Mouse dorsal root ganglia cultures

Embryonic DRG spot cultures were prepared as described by Gerdts *et al.* (*Gerdts et al., 2013*). Briefly, DRGs were cultured from embryonic day 12.5–13.5 CD1 (Charles River) or SARM1$^{-/-}$ mice (*Szretter et al., 2009*) on plates coated with poly-D-Lysine and laminin. Neurobasal culture medium (Invitrogen, Carlsbad, CA) was supplemented with 2% B27 (Invitrogen), 50 ng/mL nerve growth factor (Harlan Laboratories, Indianapolis, IN), and 1 $\mu$M 5-Fluoro-2'-deoxyuridine (Sigma) and 1 $\mu$M uridine (Sigma). All experiments were performed at DIV 8–9.

### Lentivirus transduction

Lentiviruses were generated as previously described (*Araki et al., 2004*). Virus was added to cultures at DIV2. We achieve ~100% transduction efficiency into DRG neurons. All cultures were treated with lentivirus expressing Bcl-XL to suppress non-specific shRNA toxicity. Bcl-XL overexpression does not influence the SARM1 Wallerian degeneration program or axon degeneration induced by dimerization of the SARM1-TIR domains (*Gerdts et al., 2013*; *Vohra et al., 2010*; *Yang et al., 2015*). Plasmids used in this study: NMNAT2-myc-6xHis (NMNAT2-myc; *Babetto et al., 2013*), SCG10-AA (*Shin et al., 2012*), shMKK4-2 (Sigma; TRCN0000345130, primary MKK4 shRNA used in this study), shMKK4-1 (Sigma; TRCN000025264), shMKK7 (Sigma; TRCN0000012609), shSARM1 (*Gerdts et al., 2013*), Fkbp$^{F36V}$-sTIR-Cerulean (*Gerdts et al., 2015*), CytNMNAT1 (*Sasaki and Milbrandt, 2010*), and FCIV for vector controls (*Araki et al., 2004*). Expression of constructs was confirmed by green fluorescence in cells or by western blot analysis. Multiple shRNAs were validated for MKK4.

### CRISPR/gRNA

For CRISPR/gRNA studies, Cas9 knock-in mice from Jackson Laboratory (Strain# 024857) (*Platt et al., 2014*) were bred to mice expressing CRE recombinase under control of mouse beta-Actin promoter. From this breeding, E13.5 embryos were dissected and embryonic DRGs seeded and cultured as described above. For analysis of NMNAT2 protein levels after depletion of MKK4/7, DRGs were infected with lentivirus on DIV1. Cells were lysed on DIV 8 and cell extracts analyzed by western immunoblotting. For suppression of axon protection experiments, DRGs were infected with Bcl-XL and shRNAs on DIV 3 and gRNA to *Nmnat2* or scrambled control on DIV 4. Axons were not fragmented spontaneously with gRNA to *Nmnat2* during the course of the experiment. Scrambled gRNAs (CGTCGCCGGCGAATTGACGG and CGCGGCAGCCGGTAGCTATG), *Map2k4* (MKK4)

gRNAs (TTGTTTTACAGGGCGACTGT and TTTGTAAAACTTATCGAACG), *Map2k7* (MKK7) gRNAs (TTGCAGCAAATGCGGCGCT and CCAAAGCACTGAACGATGTA), *Nmnat2* gRNA (GGAGCCCACC TGTTTTCCGT). gRNAs were designed with help from the Genome Engineering and iPSC center. gRNA sequences were cloned into LentiGuide-puro plasmid (Addgene # 52963).

## qRT-PCR

Total RNA was purified from DRG cultures using phenol-chloroform extraction using Trizol (Invitrogen). RNA was retro-transcribed using qScript cDNA supermix (Quanta Biosciences, Beverly, MA). Quantitative real-time PCR was performed using a SYBR green-based assay on a 7900 HT sequence detection system (Applied Biosystems) and quantified using delta-CT, with *Gapdh* as an internal control. Primers include: *Nmnat2* Fwd: 5' GACCGAGACCACAAAGACCC, NMNAT2 Rev: 5' CCCTGGC TCTCTCGAACATC, *Stmn2* (SCG10) Fwd: 5' TGCGTGCACATCCCTACAAT, *Stmn2* (SCG10) Rev: 5' CTGCTTCACCTCCATGTCGT, *Gapdh* Fwd: 5' TGTGAACGGATTTGGCCGTA, and *Gapdh* Rev: 5' ACTGTGCCGTTGAATTTGCC.

## Drosophila

*Drosophila melanogaster* were raised on standard fly food at 25°C. The following strains were used in this study: Canton S (wild-type), *hiw*^AN (*Wu et al., 2005*), UAS-*bsk*^DN (*Weber et al., 2000*),*OK6*-Gal4 (motoneuron specific; *Aberle et al., 2002*), m12-Gal4 (single-axon resolution; *Ritzenthaler et al., 2000*), UAS-HA-dNMNAT (*Zhai et al., 2006*), and the following RNAi lines from the Bloomington Trip collection (Bsk, 53310 and 57035). For the nerve crush assay, segmental nerves of third instar larvae were visualized through the cuticle under a dissection microscope. Larvae were immobilized in a cold dish and the segmental nerves were pinched tightly through the cuticle for five seconds with Dumostar number five forceps. After injury, the larva were transferred to a grape plate and kept alive for varying periods of time at 25°C before dissection. This assay was adapted from (*Xiong et al., 2010*). Axons were labeled with GFP expressed by the *M12-Gal4* driver and nerves were labeled with α-HRP.

## NAD$^+$ and ATP measurements

Dense spot cultures derived from four embryos were cultured in a 24-well plate. For NAD$^+$ measurement after axotomy, axons were cut at the indicated timepoints, cell bodies were removed using forceps, and NAD$^+$ was collected from the remaining neurites. NAD$^+$ was collected from whole cell for SARM1-TIR-mediated NAD$^+$ depletion experiments. Wells were rinsed twice with cold saline and metabolites were collected in 150 μL 1 M perchloric acid (HClO4) on ice for five minutes. After spinning for five minutes, the supernatant was transferred to a new tube, 20 μL 3M potassium carbonate (K$_2$CO$_3$) was added. The supernatant was loaded for HPLC after a five minute spin as described (*Gerdts et al., 2015*). NAD$^+$ measurements were normalized to the time zero controls for each condition.

## Axotomy

DRG spot cultures were axotomized using a microscalpel and images of distal axons were taken at indicated timepoints. Axon images for gRNA and SARM1-TIR experiments were acquired using an Operetta high-content imaging system (PerkinElmer, Waltham, MA). For quantification of axon degeneration, images were analyzed using ImageJ software with the Degeneration Index algorithm, which computes an unbiased degeneration score ranging from 0–1.0, with 1.0 being total fragmentation (*Sasaki et al., 2009b*). In this study, a score of about 0.3 indicates an approximate transition from a morphologically intact to a blebbing axon. Each technical replicate included 2–3 wells per condition and experiments were repeated for at least three independent biological replicates.

## Western blot

For axon-only western blot, dense spot cultures were plated using 4 embryos per 24-well plate. A microscalpel was used to remove cell bodies. Axon-only lysate was collected from 4–6 wells. Samples were lysed on ice using Laemmli buffer with protease inhibitor cocktail (Roche #11836170001).

A standard western blot procedure was followed. For quantification, bands were normalized to HSP90 or Tuj1 as loading controls. For half-life quantification, data were fit to a curve.

## Immunofluorescence

DRGs were plated in spot cultures in 8-well Permanox slides (Lab-Tek, Thermo Scientific). Neurons were fixed in 4% paraformaldehyde, washed, and blocked as described (*Shin et al., 2012*). NMNAT2-myc was detected with anti-myc tag antibody (9B11; CST #2276), and neurons were co-stained with anti-$\beta$3 tubulin. Larval filet preps of third instar larva were fixed in 4% paraformaldehyde for 20 min at room temperature. Blocking and staining were performed in PBS + 0.1% Triton X-100 + 5% goat serum. All *Drosophila* samples were mounted and imaged in 70% glycerol containing Vectashield. Samples were imaged on a Leica DMI 4000B Confocal microscope using 40x or 60x oil objectives. Images are maximal Z-projections of confocal stacks. Samples for each experiment were processed simultaneously with identical confocal gain settings. Intensity of NMNAT2-myc and SCG10 in DRGs was quantified using ImageJ by determining the mean grey value within the total axonal area as defined by Tuj1 staining with background subtracted. A similar strategy was employed for intensity of HA-dNMNAT, with HRP labeling total axonal area.

## Statistical analysis

One-way ANOVA comparison with Tukey post-hoc was used to test for statistical significance. All data are presented as mean ± SEM. Data analysis was performed in GraphPad Prism.

## Acknowledgements

We thank T Fahrner, S Johnson, P Breton, and X Sun for technical assistance and members of the DiAntonio and Milbrandt labs for fruitful discussions.

---

## Additional information

### Funding

| Funder | Grant reference number | Author |
| --- | --- | --- |
| National Institute of Neurological Disorders and Stroke | NS087632 | Jeffrey Milbrandt Aaron DiAntonio |
| Muscular Dystrophy Association | MDA349925 | Aaron DiAntonio |
| Muscular Dystrophy Association | MDA 344513 | Daniel W Summers |
| National Institute of Neurological Disorders and Stroke | NS065053 | Aaron DiAntonio |

The funders had no role in study design, data collection and interpretation, or the decision to submit the work for publication.

### Author contributions

LJW, Conceptualization, Data curation, Formal analysis, Investigation, Writing—original draft, Writing—review and editing; DWS, Data curation, Formal analysis, Funding acquisition, Investigation, Writing—review and editing; YS, EJB, Data curation, Formal analysis, Investigation, Writing—review and editing; JM, Conceptualization, Resources, Supervision, Funding acquisition, Writing—review and editing; AD, Conceptualization, Resources, Supervision, Funding acquisition, Writing—original draft, Writing—review and editing

### Author ORCIDs

Lauren J Walker, http://orcid.org/0000-0003-3844-4275
Yo Sasaki, http://orcid.org/0000-0003-0024-0031
Aaron DiAntonio, http://orcid.org/0000-0002-7262-0968

### Ethics

Animal experimentation: This study was performed in strict accordance with the recommendations in the Guide for the Care and Use of Laboratory Animals of the National Institutes of Health. All of the animals were handled according to approved institutional animal care and use committee (IACUC) protocols (#A-3381- 01) at Washington University in St. Louis. The protocol was approved by the Committee on the Ethics of Animal Experiments of the University of Minnesota (Permit Number: 2015043).

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
