## [Decision Letter]

[Editors’ note: a previous version of this study was rejected after peer review, but the authors submitted for reconsideration. The first decision letter after peer review is shown below.]

Thank you for submitting your work entitled "MAPK Signaling Promotes Axonal Degeneration by Speeding the Turnover of the Axonal Maintenance Factor NMNAT2" for consideration by *eLife*. Your article has been reviewed by three peer reviewers, and the evaluation has been overseen by a Reviewing Editor and Marianne Bronner as the Senior Editor. The reviewers have opted to remain anonymous.

Our decision has been reached after consultation between the reviewers. Based on these discussions and the individual reviews below, we regret to inform you that your work will not be considered further for publication in *eLife*.

Overall, the reviewers were positive both in their reviews and in their online discussion. As one reviewer commented in our discussion, your study on the relationship between injury, MAPK signaling, NMNAT2, and NAD^+^ level alterations constitutes "a significant data set that will contribute to the eventual full understanding of degenerative pathways".

The reviewers recognize that your results differ from those in Yang et al., from the Tessier-Lavigne group but in no way dismiss your results. Rather, they offer suggestions, albeit requiring additional experiments, to parse out MKK4/7 control of SARM activation that in turn regulates the neuroprotective elements NMNAT2 and SCG10:

Although only Reviewer 1 noted this in the reviews below, all three agreed in our discussion that the Bcl-XL overexpression, to enhance neuron viability after transfection with shRNA constructs, is one aspect that could contribute to these differences. Bcl-XL overexpression should be explained and discussed in the text and, ideally, tested.

The epistasis experiments are welcome, but the reviewers called for analysis of activation of individual components. They also agree that it would be important to test the effects of reducing JNK activity.

All three reviewers were concerned about the conclusion that the MAPK pathway regulates degradation: They ask for clarification on the degradation and turnover analyses that lead to the conclusion that MKK4/7 promote axonal turnover of NMNAT2 and SCG10 post-injury. Reviewer 2 indicates that the lower rate of protein degradation could result from the degradation components being saturated with the high levels of NMNAT2 and SC10 seen in the MKK4/7/ depleted condition. Reviewer 3 asks whether this is due to axonal turnover depending on concurrent MAP kinase activity, or increased axonal loading overwhelming the normal course of degradation. A suggested experiment would involve increasing the levels of NMNAT2 and SCG10 in cells (to levels comparable to those seen with MKK4/7/ depletion) and show that the rate of degradation is unaltered.

Reviewer 3 asked for the FRET experiment to directly readout SARM, but believes that this would be challenging, and so we will not hold you to perform this experiment.

Thus, based on consultation among the three reviewers with the Reviewing Editor on these aspects and the individual reviews below, several asking for additional analyses, we regret to inform you that your work as it stands will not be considered for publication in *eLife* at this time. We would be happy to receive a new submission if you were to address the reviewers' concerns.

*Reviewer #1:*

The mechanism of axon degeneration is an area of great interest and intense investigation. Several highly conserved components involved in axotomy induced injury have been identified, including a critical prodegenerative component SARM, a distinctive MAPK pathway comprised of DLK/MEKK4/JNKs, and an anti-degenerative metabolic enzyme NMNAT. Together these molecules alter the metabolic state of axons, and regulate axonal degeneration. Additional components, including ubiquitinating enzymes, Bcl2 family members, caspases, and calpain have all been tied to regulation of axon degeneration. Here Walker et al. try to develop a cohesive view of these multiple components, and how they work together. The findings presented contradict a recent paper in the field. While that study concluded that SARM is upstream of the MAPK cascade, the present study reaches the opposite conclusion, namely that MAPK regulates degradation of NMNAT2 (and other components). Thus, in reviewing the paper, I tried to consider both how the current study differs from Yang et al., and also how well the current model is explored. A major problem (with both studies) is the push to place all these components in a single linear pathway. The impetus to place all the components in a linear pathway may have oversimplified this complex process.

1) A major methodologic difference with the Yang et al. report is that the current study expresses Bcl-XL in all cultures in which shRNA knock down is performed. (Subsection “Lentivirus Transduction”, "All cultures were treated with lentivirus expressing Bcl-XL to suppress non-specific shRNA toxicity). The authors note that this enhances the viability of neurons transfected with shRNA constructs. However, the Tessier-Lavigne group has reported that bclxl can promote axonal survival and prevent degeneration. Therefore, this overexpression could definitely alter the results throughout the study (all experimental figures with the exception of Figure 4 seem to be affected by this issue).

2) The results suggest that MAPK is needed for activation of SARM1, but once activated SARM functions independently of MAPK. IF this is true, then genetic epistasis experiments showing that MAPK is not needed for the activity of a constitutively active SARM, do not indicate that MAPK must be downstream of SARM. I believe there could indeed be a feedback loop. The authors considered, and rejected this possibility. The rejection of this model was based on evidence that MAPK regulates NMNAT2 in the absence of SARM. Again, this result is only problematic if the authors feel that there is a single input into this MAPK cascade.

3) In Figure 1, NAD and ATP are measured in axonal lysates following axotomy, but are instead measured in lysates of the whole cells after SARM dimerization. It is possible that MKK4/7 only alters NAD and ATP locally in axons, but does so both in response to axotomy or to SARM activation, Therefore the conclusion that MKK4/7 only alter NAD and ATP following axotomy and not following SARM activation is somewhat problematic.

4) I think that genetic epistatis experiments are very useful, but it is also important to examine directly activation of individual components. Does activated SARM stimulate the DLK/Mkk4/JNK pathway? To address this question, the authors should look at phosphorylation or catalytic activity of pathway components as an indicator that the MAPK cascade is activated. This is particularly important since the genetic epistasis experiments are not all consistent, and it is likely that many of the pathway components have multiple functions.

*Reviewer #2:*

This manuscript by Walker and colleagues provides interesting mechanistic insights into the pathway of Wallerian degeneration, which has made significant advances in recent years. Specifically, this manuscript elucidates the order of events that have been recently shown to be key mediators of this pathway, namely-MAPK, NMNAT, and Sarm1. The authors here show that MKK4/7 are important for the loss of NAD^+^ in axons in response to axotomy and find that MKK4/7 depletion does not prevent the loss of NAD^+^ in neurons where Sarm1 is directly activated. These findings are interesting because MKK4/7 was previously shown to function after the point of Sarm1 activation to mediate axon degeneration. However, a major point of this manuscript is MKK4/7 functions before the point of Sarm1 activation in this pathway.

The authors follow up this finding by identifying the role of MKK4/7 in regulating the levels of NMNAT2 and SCG10, both previously implicated in protecting against Wallerian degeneration. Loss of MKK4/7 leads to increase levels of NMNAT2 and SCG10, which maintain axon survival.

Overall, these are interesting findings. However, additional experiments are needed to strengthen the conclusions. Also, the current title is not fully supported with the data shown. The data mainly show that MKK4/7 acts upstream of Sarm1 activation and mediates axon degeneration by regulating levels of NMNAT2 and SCG10.

1) The differences in results between this manuscript and the Yang 2015 paper are significant and there does not seem to be any straightforward explanation for this. The Yang 2015 paper shows that both MKK4/7 depletion and JNK1/2/3 depletion prevented active Sarm1-induced axon degeneration. Since the results in his manuscript show that MKK4/7 depletion did not block axon degeneration in the context of direct Sarm activation, the authors should also examine whether similar results are obtained with JNK depletion (or inhibition).

2) In Figure 3, it appears as though MKK4/7 depletion may also decrease Sarm1 levels. Can the authors comment on this observation? If this is indeed the case, then decreasing levels of Sarm1 could be another mechanism by which MKK4/7 depletion may protect axon degeneration.

3) In Figure 5, the results showing that MKK4/7 depletion decreases the rate of NMNAT2 and SCG10 degradation is not convincing. For example, the lower rate of protein degradation could be because the degradation components are saturated with the high levels of NMNAT2 and SC10 seen in the MKK4/7/ depleted condition. To eliminate that possibility, one would need to increase the levels of NMNAT2 and SCG10 in cells (to levels comparable to those seen with MKK4/7/ depletion) and show that the rate of degradation is unaltered.

4) Have the authors considered increased transcription or translation as factors that could result in the increased accumulation of NMNAT2 or SCG10 in MKK4/7 depleted conditions? Since the title of this manuscript seems to imply turnover as the main mechanism, they need to be more rigorous to not only show convincing degradation data but also to eliminate the other possibilities.

5) In the third paragraph of the subsection “MKK4/7 are necessary for NAD^+^ loss and axon degeneration after axotomy but not in response to constitutively active SARM1”, the authors conclude that MKK4/7 are "required" for NAD depletion after axotomy. Perhaps it is more accurate to state that MKK4/7 is important for NAD depletion after axotomy (since NAD levels are decreased nearly 50% even in the sh4/7 condition; Figure 1).

*Reviewer #3:*

This paper revises the current understanding of the relationship between injury, MAPK signaling, NMNAT2, and NAD^+^. These are interesting and significant findings given the prominent role of these proteins in axon degeneration. I have a few suggestions before this manuscript would be appropriate for *eLife*.

a) Although the epistasis is convincing, in general a direct readout of SARM would be very powerful. This would allow testing whether activating or inactivating MAP kinase signaling is sufficient to activate SARM, independently of axon injury. Perhaps this could be done with a FRET reporter for SARM conformation.

b) The experiments in cultured cells and flies suggest that MAP kinase signaling is active even in uninjured neurons, and acts to limit levels of axon survival factors. This is a surprising result that needs more discussion. Can MAP kinase signaling be blocked acutely? How quick is the resulting accumulation of NMNAT2 and SCG10?

c) An important conclusion is that MKK4/7 promote axonal turnover of NMNAT2 and SCG10 post-injury. However, in these experiments the initial amount of NMNAT2 and SCG10 is greatly increased. Thus, it is impossible to know whether axonal turnover really depends on concurrent MAP kinase activity, or whether increased axonal loading overwhelms the normal course of degradation. Perhaps these proteins could be overexpressed to mimic what happens in the MKK4/7 knockdown, and then turnover after injury could be measured.

d) In the final experiment, NMNAT2 is shown to be downstream of MKK4/7, but not SARM. I have concerns about the interpretation. For NMNAT2, the data do show that "NMNAT2 is required for the axonal protection afforded by MKK4/7 knockdown", as stated, but not that "the protective effect of inhibiting MAPK signaling is conferred by elevated levels of NMNAT2" (subsection “MAPK signaling promotes axonal degeneration via regulation of survival factors”, first paragraph). In fact, the proposed model also includes SGC10, and there may be other protective effectors as well.

[Editors’ note: what now follows is the decision letter after the authors submitted for further consideration.]

Thank you for submitting your article "MAPK Signaling Promotes Axonal Degeneration by Speeding the Turnover of the Axonal Maintenance Factor NMNAT2" for consideration by *eLife*. Your article has been reviewed by three peer reviewers, and the evaluation has been overseen by a Reviewing Editor and Marianne Bronner as the Senior Editor. The reviewers have opted to remain anonymous.

The reviewers have discussed the reviews with one another and the Reviewing Editor has drafted this decision to help you prepare a revised submission.

Summary:

All three reviewers are satisfied with the amendments and agree that the manuscript is strengthened. They have two remaining sets of recommendations.

Essential revisions:

Reviewer 3 asks that some of the key data be included in the revised manuscript, either in the main or figure supplements. This reviewer believes that including the following data in figures is important, especially as some of the data directly address and sometimes contradict the published data from Yang et al.

a) You repeated the SARM1-TIR experiment using JNK inhibitor and found that, like MKK4/7 depletion, JNKi did not block axon degeneration induced by direct SARM1 activation. In particular, these data should be in the main figures.

b) You report that NMNAT2 and SCG10 transcripts were not elevated upon depletion of MKK4/7 by RT-PCR. Based on this result, and the turnover data, you conclude that MAPKs influence the turnover rate of NMNAT2 and SCG10. Since this result is important for the overall conclusion of the manuscript, it would be important to also show it in a figure.

---

## [Author Response]

[Editors’ note: the author responses to the first round of peer review follow.]

*The reviewers recognize that your results differ from those in Yang et al., from the Tessier-Lavigne group but in no way dismiss your results. Rather, they offer suggestions, albeit requiring additional experiments, to parse out MKK4/7 control of SARM activation that in turn regulates the neuroprotective elements NMNAT2 and SCG10:*

*Although only Reviewer 1 noted this in the reviews below, all three agreed in our discussion that the Bcl-XL overexpression, to enhance neuron viability after transfection with shRNA constructs, is one aspect that could contribute to these differences. Bcl-XL overexpression should be explained and discussed in the text and, ideally, tested.*

We appreciate the reviewer’s suggestion and apologize for not more clearly describing all of the prior published work demonstrating that Bcl-XL overexpression is not the explanation for the difference between the findings from the two groups. Both the Tessier-Lavigne group and our groups have published that Bcl-XL overexpression *does not influence* the SARM1 Wallerian degeneration program. Indeed, the one point of experimental (rather than interpretational) disagreement between our work and the work of Yang et al. relates to the degeneration induced by dimerizing the SARM1-TIR domain. Importantly, Yang et al. state that this degeneration *is not influenced* by overexpression of Bcl-XL. In the Yang et al. paper, the Tessier-Lavigne group finds that “Treatment of sensory neurons expressing FKBP(F36V)-TIR with AP20187 induced degeneration of distal axons (Figure 4), which appears independent of the apoptotic pathway since it was not affected by genetic deletion of caspase-9 or *overexpression of Bcl-xl*(data not shown)” (Yang et al., 2015). We also previously demonstrated that Bcl-XL overexpression does not affect axon degeneration induced by activated SARM 1 (Gerdts et al., 2013, Figure 6). We have also previously published that overexpression of Bcl-XL does not influence the progression of axotomy-induced axon degeneration, which is dependent on full-length SARM1 (Vohra et al., 2010, Figure 1). Reviewer one notes that the Tessier-Lavigne group has demonstrated an important role for Bcl-XL in inhibiting axon degeneration, but that is in the paradigm of NGF-deprivation not axotomy. The Tessier-Lavigne group has shown that NGF-deprivation activates a different molecular pathway than is used in the SARM1-dependent Wallerian degeneration pathway (Simon et al., 2012, 2016). We have published a similar finding for the role of Bcl-XL in NGF-deprivation induced axon degeneration (Vohra et al., 2010). Finally, the contradiction between our work and the work of Yang et al. would not be explained by the expression of Bcl-XL even if a role for Bcl-XL had not been previously excluded by the publications above. We find that knockdown of MKK4/7 does not block SARM1-TIR-induced axon degeneration, while Yang et al. does see protection. The *lack* of protection in our system could not be explained by the expression of a protective factor such as Bcl-XL.

We have now added additional explanation and references in the section where we discuss the Bcl-XL overexpression.

*The epistasis experiments are welcome, but the reviewers called for analysis of activation of individual components. They also agree that it would be important to test the effects of reducing JNK activity.*

We have addressed these issues via a series of experiments. We asked about MAPK activation by testing whether dimerization of the SARM1-TIR domain is sufficient to activate the MAPK pathway as described by Yang et al. We now show in Figure 2—figure supplement 1 that we also observe activation of MKK4 following SARM1-TIR dimerization. Hence, we do not disagree with Yang et al. that forced dimerization of SARM1-TIR *can* induce activation of the MAPK stress pathway. We think this reconciles some of the apparent contradiction between our work and the Yang et al. study. What we attempt to do throughout the paper, however, is to assess whether this activation is *the functionally relevant activation*. We go on to find a downstream function of MAPK signaling (increased levels of NMNAT2 and SCG10), we show that this function is *not downstream of SARM1* because it occurs in SARM1 KO neurons, and demonstrate that this function is necessary for the protection afforded by loss of MAPK signaling in our epistasis experiments. We do now expand our model to acknowledge this additional MAPK activation, and while we have no evidence that it is functionally relevant, we do discuss how this activation could serve to promote axon degeneration.

We also perform additional experiments with JNK inhibition as requested. We now show that treatment of DRGs with JNK inhibitor is sufficient to increase the levels of NMNAT2 and SCG10 within two hours (new Figure 3) and demonstrate that acute treatment with JNK inhibitor slows the turnover rate of NMNAT2 and SCG10 (Figure 5—figure supplement 1). Finally, we show that JNK inhibition, like MKK4/7 inhibition, does not block SARM-TIR induced axon degeneration. All of these findings are fully consistent with our prior results with MKK4/7 inhibition and strengthen the evidence for our model.

*All three reviewers were concerned about the conclusion that the MAPK pathway regulates degradation: They ask for clarification on the degradation and turnover analyses, that lead to the conclusion that MKK4/7 promote axonal turnover of NMNAT2 and SCG10 post-injury. Reviewer 2 indicates that the lower rate of protein degradation could result from the degradation components being saturated with the high levels of NMNAT2 and SC10 seen in the MKK4/7/ depleted condition. Reviewer 3 asks whether this is due to axonal turnover depending on concurrent MAP kinase activity, or increased axonal loading overwhelming the normal course of degradation. A suggested experiment would involve increasing the levels of NMNAT2 and SCG10 in cells (to levels comparable to those seen with MKK4/7/ depletion) and show that the rate of degradation is unaltered.*

We thank the reviewers for the suggestions. First, we demonstrate that acute treatment with JNK inhibitor slows the turnover rate of survival factors before there is time for a significant increase in the levels of NMNAT2 and SCG10 (Figure 5—figure supplement 1). This is very strong evidence that inhibiting this MAPK pathway is controlling NMNAT2 and SCG10 levels by slowing their turnover.

Second, we performed the experiment requested by the reviewers. We find that when we simultaneously overexpress NMNAT2 and SCG10 (to mimic the effects of MAPK inhibition) their turnover rate is indistinguishable from their turnover rate when wild type levels of protein are present (Figure 5—figure supplement 2). Hence, we have ruled out the hypothesis that the increased levels of survival factors have saturated the protein degradation machinery. Together, these two experiments strongly support our hypothesis that the MAPK pathway controls the levels of these survival factors by regulating their turnover rate.

*Reviewer 3 asked for the FRET experiment to directly readout SARM, but believes that this would be challenging, and so we will not hold you to perform this experiment.*

We appreciate that the reviewers recognize that this would be beyond the scope of this proposal.

*Reviewer #1:*

*[…] 1) A major methodologic difference with the Yang et al. report is that the current study expresses bclxl in all cultures in which shRNA knock down is performed. (Subsection “Lentivirus Transduction”, "All cultures were treated with lentivirus expressing Bcl-XL to suppress non-specific shRNA toxicity). The authors note that this enhances the viability of neurons transfected with shRNA constructs. However, the Tessier-Lavigne group has reported that bclxl can promote axonal survival and prevent degeneration. Therefore, this overexpression could definitely alter the results throughout the study (all experimental figures with the exception of Figure 4 seem to be affected by this issue).*

We addressed this point in detail above, and have included clarification of this topic so that readers will not be confused.

*2) The results suggest that MAPK is needed for activation of SARM1, but once activated SARM functions independently of MAPK. IF this is true, then genetic epistasis experiments showing that MAPK is not needed for the activity of a constitutively active SARM, do not indicate that MAPK must be downstream of SARM. I believe there could indeed be a feedback loop. The authors considered, and rejected this possibility. The rejection of this model was based on evidence that MAPK regulates NMNAT2 in the absence of SARM. Again, this result is only problematic if the authors feel that there is a single input into this MAPK cascade.*

All of our functional data are consistent with a linear model in which the MAPK pathway negatively regulates NMNAT2 levels to control SARM1 activation and subsequent NAD^+^ depletion. Because this simple model can explain all of the functional data, this is our preferred model. However, we certainly acknowledge that biology is complicated, and that additional regulatory mechanisms likely exist. We now discuss that SARM1-dependent activation of the MAPK pathway, although not necessary for the regulation of NMNAT2 and SCG10, could play an additional regulatory role in fine-tuning the process of axon degeneration.

*3) In Figure 1, NAD and ATP are measured in axonal lysates following axotomy, but are instead measured in lysates of the whole cells after SARM dimerization. It is possible that MKK4/7 only alters NAD and ATP locally in axons, but does so both in response to axotomy or to SARM activation, Therefore the conclusion that MKK4/7 only alter NAD and ATP following axotomy and not following SARM activation is somewhat problematic.*

We appreciate the suggestion and performed the suggested experiment. In our original submission we showed that MKK4/7 does not influence total *neuronal* NAD^+^ following SARM1-TIR dimerization. We now demonstrate that MKK4/7 do not influence local *axonal* NAD^+^ following SARM1- TIR dimerization. This finding is fully consistent with our prior result.

*4) I think that genetic epistatis experiments are very useful, but it is also important to examine directly activation of individual components. Does activated SARM stimulate the DLK/Mkk4/JNK pathway? To address this question, the authors should look at phosphorylation or catalytic activity of pathway components as an indicator that the MAPK cascade is activated. This is particularly important since the genetic epistasis experiments are not all consistent, and it is likely that many of the pathway components have multiple functions.*

We have discussed this above. We now confirm the finding of Yang et al. that activation SARM1 can activate the MAPK stress pathway. We now incorporate this finding into our Discussion.

*Reviewer #2:*

*[…] 1) The differences in results between this manuscript and the Yang 2015 paper are significant and there does not seem to be any straightforward explanation for this. The Yang 2015 paper shows that both MKK4/7 depletion and JNK1/2/3 depletion prevented active Sarm1-induced axon degeneration. Since the results in his manuscript show that MKK4/7 depletion did not block axon degeneration in the context of direct Sarm activation, the authors should also examine whether similar results are obtained with JNK depletion (or inhibition).*

We repeated the SARM1-TIR experiment using JNK inhibitor and find that, like MKK4/7 depletion, JNKi does not block axon degeneration induced by direct SARM1 activation. This result has been incorporated into the Results section and fully supports our model.

*2) In Figure 3, it appears as though MKK4/7 depletion may also decrease Sarm1 levels. Can the authors comment on this observation? If this is indeed the case, then decreasing levels of Sarm1 could be another mechanism by which MKK4/7 depletion may protect axon degeneration.*

We thank the reviewer for this observation. After quantifying 6 blots, we find that there is no significant change in SARM1 levels upon depletion of MKK4/7.

*3) In Figure 5, the results showing that MKK4/7 depletion decreases the rate of NMNAT2 and SCG10 degradation is not convincing. For example, the lower rate of protein degradation could be because the degradation components are saturated with the high levels of NMNAT2 and SC10 seen in the MKK4/7/ depleted condition. To eliminate that possibility, one would need to increase the levels of NMNAT2 and SCG10 in cells (to levels comparable to those seen with MKK4/7/ depletion) and show that the rate of degradation is unaltered.*

We appreciate the experimental suggestion. To address this concern, we have taken two approaches. First, as the reviewer suggested, we have overexpressed NMNAT2 and SCG10 to levels approximately 5 to 8-fold above endogenous levels (Figure 5—figure supplement 2). We find that the turnover rate of the overexpressed protein is comparable to endogenous conditions. Furthermore, we find that acute treatment with JNK inhibitor, just 30 minutes prior to cycloheximide turnover experiments, is sufficient to increase the half-life of NMNAT2 (Figure 5—figure supplement 1). In this experimental paradigm, there is not enough time for extensive buildup of NMNAT2 or other proteins to impact the degradation machinery by their abundance, and instead argues that MAPKs are directly influencing protein turnover rate.

*4) Have the authors considered increased transcription or translation as factors that could result in the increased accumulation of NMNAT2 or SCG10 in MKK4/7 depleted conditions? Since the title of this manuscript seems to imply turnover as the main mechanism, they need to be more rigorous to not only show convincing degradation data but also to eliminate the other possibilities.*

We find that NMNAT2 and SCG10 transcripts are not elevated upon depletion of MKK4/7 by rt-PCR (measured transcripts from 3 independent experiments). Thus, taken together with the additional turnover data from point 3 above, we are confident to conclude that MAPKs influence the turnover rate of NMNAT2 and SCG10. As such, we think it would be inappropriate to remove ‘turnover’ from the title.

*5) In the third paragraph of the subsection “MKK4/7 are necessary for NAD^+^ loss and axon degeneration after axotomy but not in response to constitutively active SARM1”, the authors conclude that MKK4/7 are "required" for NAD depletion after axotomy. Perhaps it is more accurate to state that MKK4/7 is important for NAD depletion after axotomy (since NAD levels are decreased nearly 50% even in the sh4/7 condition; Figure 1).*

We have changed the text to reflect this change.

*Reviewer #3:*

*[…] a) Although the epistasis is convincing, in general a direct readout of SARM would be very powerful. This would allow testing whether activating or inactivating MAP kinase signaling is sufficient to activate SARM, independently of axon injury. Perhaps this could be done with a FRET reporter for SARM conformation.*

We agree that a FRET reporter for SARM1 activation would have numerous experimental applications. We appreciate that the reviewers find this to be beyond the scope of this study.

*b) The experiments in cultured cells and flies suggest that MAP kinase signaling is active even in uninjured neurons, and acts to limit levels of axon survival factors. This is a surprising result that needs more discussion. Can MAP kinase signaling be blocked acutely? How quick is the resulting accumulation of NMNAT2 and SCG10?*

We include a time course of NMNAT2 and SCG10 levels after treatment with JNK inhibitor in a new Figure 3. We observe an increase of NMNAT2 and SCG10 levels within two hours of JNK inhibitor treatment.

*c) An important conclusion is that MKK4/7 promote axonal turnover of NMNAT2 and SCG10 post-injury. However, in these experiments the initial amount of NMNAT2 and SCG10 is greatly increased. Thus, it is impossible to know whether axonal turnover really depends on concurrent MAP kinase activity, or whether increased axonal loading overwhelms the normal course of degradation. Perhaps these proteins could be overexpressed to mimic what happens in the MKK4/7 knockdown, and then turnover after injury could be measured.*

This was the same suggestion made by reviewer 2 and is addressed above.

*d) In the final experiment, NMNAT2 is shown to be downstream of MKK4/7, but not SARM. I have concerns about the interpretation. For NMNAT2, the data do show that "NMNAT2 is required for the axonal protection afforded by MKK4/7 knockdown", as stated, but not that "the protective effect of inhibiting MAPK signaling is conferred by elevated levels of NMNAT2" (subsection “MAPK signaling promotes axonal degeneration via regulation of survival factors”, first paragraph). In fact, the proposed model also includes SGC10, and there may be other protective effectors as well.*

While our data demonstrate that NMNAT2 is required for the protective effect afforded by MKK4/7 knockdown, we have modified the text to note that SCG10 and other protective factors may also play a role in MAPK-mediate axon degeneration.

[Editors' note: the author responses to the re-review follow.]

*Essential revisions:*

*Reviewer 3 asks that some of the key data be included in the revised manuscript, either in the main or figure supplements. This reviewer believes that including the following data in figures is important, especially as some of the data directly address and sometimes contradict the published data from Yang et al.*

*a) You repeated the SARM1-TIR experiment using JNK inhibitor and found that, like MKK4/7 depletion, JNKi did not block axon degeneration induced by direct SARM1 activation. In particular, these data should be in the main figures.*

This is now included as Figure 2.

*b) You report that NMNAT2 and SCG10 transcripts were not elevated upon depletion of MKK4/7 by RT-PCR. Based on this result, and the turnover data, you conclude that MAPKs influence the turnover rate of NMNAT2 and SCG10. Since this result is important for the overall conclusion of the manuscript, it would be important to also show it in a figure.*

This is now included as Figure 5.